# Control of nuclear size by osmotic forces in *Schizosaccharomyces pombe*

**Joël Lemière[1], Paula Real-Calderon[1,2], Liam J Holt[3], Thomas G Fai[4]\*, Fred Chang[1]\***

[1]Department of Cell and Tissue Biology, University of California, San Francisco, San Francisco, United States; [2]Centro Andaluz de Biología del Desarrollo, Sevilla, Spain; [3]Institute for Systems Genetics, New York University Langone Health, New York, United States; [4]Department of Mathematics and Volen Center for Complex Systems, Brandeis University, Waltham, United States

**Abstract** The size of the nucleus scales robustly with cell size so that the nuclear-to-cell volume ratio (N/C ratio) is maintained during cell growth in many cell types. The mechanism responsible for this scaling remains mysterious. Previous studies have established that the N/C ratio is not determined by DNA amount but is instead influenced by factors such as nuclear envelope mechanics and nuclear transport. Here, we developed a quantitative model for nuclear size control based upon colloid osmotic pressure and tested key predictions in the fission yeast *Schizosaccharomyces pombe*. This model posits that the N/C ratio is determined by the numbers of macromolecules in the nucleoplasm and cytoplasm. Osmotic shift experiments showed that the fission yeast nucleus behaves as an ideal osmometer whose volume is primarily dictated by osmotic forces. Inhibition of nuclear export caused accumulation of macromolecules in the nucleoplasm, leading to nuclear swelling. We further demonstrated that the N/C ratio is maintained by a homeostasis mechanism based upon synthesis of macromolecules during growth. These studies demonstrate the functions of colloid osmotic pressure in intracellular organization and size control.

**\*For correspondence:**
tfai@brandeis.edu (TGF);
fred.chang@ucsf.edu (FC)

**Competing interest:** The authors declare that no competing interests exist.

## Editor's evaluation

This work offers a simple explanation to a fundamental question in cell biology: what dictates the volume of a cell and of its nucleus, focusing on yeast cells. The central message is that all this can be explained by an osmotic equilibrium, using the classical Van't Hoff's Law. The novelty resides in an effort to provide actual numbers experimentally.

## Introduction

It has been known for more than a century that the size of the nucleus scales with cell size. Since the initial observation in plants (*Strasburger, 1893*) the scaling of nuclear and cell volume has been documented across the eukaryotic domain (*Conklin, 1912*; *Gregory, 2005*; *Moore et al., 2019*; *Neumann and Nurse, 2007*). More recently, scaling was even observed for nucleoids in prokaryotes (*Gray et al., 2019*). In multicellular organisms, the nuclear-to-cell volume (N/C) ratio varies among cell types, but this ratio is generally maintained as a constant within a given cell type (*Conklin, 1912*; *Hertwig, 1903*). During cell growth, the N/C ratio is also maintained through much of the cell cycle (*Jorgensen et al., 2007*; *Neumann and Nurse, 2007*; *Willis et al., 2016*), as the nucleus grows in volume at the same rate as the cell. Abnormal N/C ratios are a hallmark of diseases such as certain cancers and are sometimes used as diagnostic criteria (*Foraker, 1954*; *Slater et al., 2005*; *Webster et al., 2009*; *Zink et al., 2004*). The N/C ratio may play an important role in regulatory mechanisms, for instance, in the mid-blastula transition in embryonic development (*Amodeo et al., 2015*; *Jevtić and Levy, 2015*).

However, despite the universal and fundamental nature of this cellular property, the mechanistic basis for nuclear size scaling remains poorly understood.

Although there is a correlation between nuclear size and amount of DNA, it is unlikely that DNA itself is the responsible scaling factor. DNA is only a minor component in the nucleus by volume; it has been estimated to occupy <1% of the nuclear volume and is many times less abundant in the nucleus than RNA (*Milo and Phillips, 2015*). Nuclear size does increase with increased ploidy in a given cell type, but generally this increase is accompanied by a similar increase in cell size (*Cavalier-Smith, 2005*; *Gregory, 2005*; *Gregory and Mable, 2005*; *Jorgensen et al., 2007*; *Robinson et al., 2018*). During the cell cycle, nuclear size continues to grow in the G2 phase even when DNA content is no longer increasing (*Jorgensen et al., 2007*; *Neumann and Nurse, 2007*). Further, through manipulating genome content in fission yeast, it has been shown that cells with DNA content ranging from 2N to 32N have a similar N/C ratio (*Neumann and Nurse, 2007*). Thus, DNA is unlikely to be the rate-limiting structural component that determines nuclear size.

Nuclear size and shape are dictated both by nuclear volume and surface area. It is clear however that nuclear volume and surface area can be uncoupled and are regulated independently. For instance, arrest of budding yeast cells in mitosis can lead to continued growth of the nuclear envelope without growth in nuclear volume, leading to misshapen nuclei and formation of nuclear envelope protrusions (*Webster et al., 2009*). Growth of the nuclear envelope may occur through the transfer of membranes from the endoplasmic reticulum or by lipid assembly at the nuclear envelope (*Blank et al., 2017*; *Hirano et al., 2020*; *Kim et al., 2007*). Studies have shown, however, that nuclear volume, not surface area, is the relevant geometric parameter that is maintained for the N/C ratio (*Cantwell and Nurse, 2019a*; *Neumann and Nurse, 2007*; *Walters et al., 2019*).

Efforts to define molecular-based control mechanisms have been largely unsuccessful. Genome-wide screens in fission yeast have demonstrated that mutants in the vast majority of genes exhibit normal N/C ratios, ruling out many possible cellular processes and molecular pathways (*Cantwell and Nurse, 2019b*; *Kume et al., 2017*). For instance, the N/C ratio is independent of cell size, shape, and number of nuclei (*Neumann and Nurse, 2007*). Screens have so far identified only a small number of genes that impact the N/C ratio, mostly related to nuclear transport or lipid synthesis (*Cantwell and Nurse, 2019b*; *Kume et al., 2017*). In vertebrate cell systems, lamins and chromatin factors have been implicated in the control of nuclear size and shape (*Edens et al., 2017*; *Levy and Heald, 2010*; *Muchir et al., 2004*). For example, depletion of lamin in *Xenopus* eggs extract resulted in a reduction of nuclear size and formation of abnormal nuclear shapes (*Newport et al., 1990*). However, as yeast lack lamins, it is unlikely that the nuclear lamins themselves represent a universal mechanism for nuclear size control.

Another potential factor in nuclear size control is osmotic pressure. Instead of a rigid structure, the nucleus may be regarded as a structure similar to a balloon whose size is dependent on the balance of pressures and membrane tension. The rounded shape of the typical nucleus suggests there may be slightly higher osmotic pressure in the nucleoplasm compared to the cytoplasm, which is balanced by the nuclear membrane tension. These pressures likely arise from macromolecular crowding forces termed 'colloid osmotic pressure', which are produced by the distinct sets of macromolecules in the nucleus and cytoplasm (*Mitchison, 2019*). The osmotic nature of the nucleus has been shown in various ways. Treatment of cells with osmotic shocks causes both the cell and nucleus to swell and shrink (*Churney, 1942*). Classic experiments demonstrated that injection of crowding agents such as polyethylene glycol into the cytoplasm cause shrinkage of the nucleus (*Harding and Feldherr, 1958*; *Harding and Feldherr, 1959*). Isolated nuclei are also responsive to osmotic shifts but the osmotic behavior depends on the molecular size of the osmolytes such that only macromolecules larger than 30 kDa will affect their volumes (*Finan et al., 2009*). In general, a rigorous quantitative assessment of the osmotic model for nuclear size control is lacking.

Here, we developed a quantitative model for nuclear size control based upon osmotic forces, using a combination of theoretical modeling and quantitative experiments. We used fission yeast as a tractable model in which cellular and nuclear volumes can be accurately measured. We propose a theoretical framework that represents the nucleus and cell as a system of nested osmometers. We show that nuclei in fission yeast behave as ideal osmometers, which allows for the direct study of the effects of osmotic pressure on nuclear volume and its responses to changes in macromolecular crowding. This osmotic model suggests a mechanism for maintenance of the N/C ratio during cell growth, as well

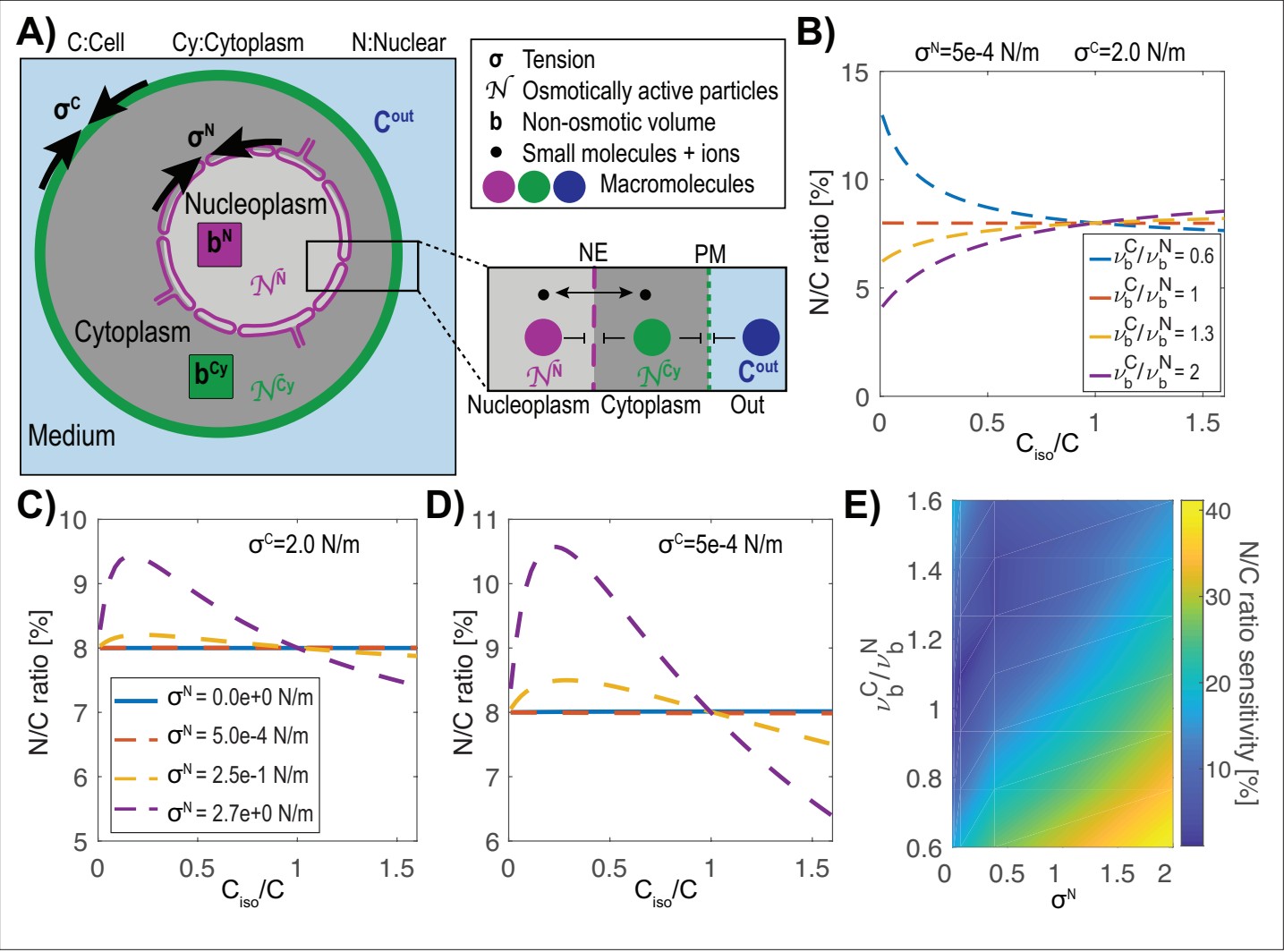

**Figure 1.** Model of the nucleus and the cell as "a vesicle within a vesicle", osmotically challenged. (**A**) Schematic of the model and parameters used in the mathematical model: membrane tension σ, non-osmotic volume b, number (**N**) of macromolecules that cannot freely cross either the cell or nuclear membranes, concentration of the buffer $C^{out}$. (**B**) Theoretical prediction of the effect of a change in the external concentration on the N/C ratio for various ratios of normalized cellular ($\nu_C$) and nuclear ($\nu_N$) non-osmotic volume values keeping the cell and nuclear membrane tensions ($\sigma^C$, $\sigma^N$) constant. (**C**) Predictions of osmotic shifts on the N/C ratio for various nuclear membrane tensions ($\sigma^N$), keeping a high cell tension ($\sigma^C$) constant. (**D**) Same as (**C**) keeping a low membrane tension. (**E**) Phase diagram of the N/C ratio sensitivity to osmotic shocks defined as [max(N/C ratio)-min(N/C ratio)] / (N/C ratio$_{isotonic}$) for various ratios of non-osmotic volumes and nuclear membrane tension.

as for homeostasis behavior that corrects an aberrant N/C ratio over time. Together, these studies provide critical quantitative support for an osmotic-based mechanism for nuclear size control.

## Results

### Model of the nucleus and a cell as two nested osmometers

We developed a quantitative model of nuclear and cell size control based on the physical mechanism of osmosis. The nucleus and the cell are represented as a system of nested osmometers, whose volumes are determined by osmotic pressure differences, membrane tensions, and non-osmotic volumes (*Figure 1A*). The cell is inflated by turgor pressure, which is defined as the osmotic pressure difference across the plasma membrane ($C^{out}$, $C^{Cy}$) balanced by the elastic wall surrounding the cell. Turgor pressure is produced largely from small molecules, such as ions and metabolites, attracting water into the cell through osmosis. The nuclear envelope is a semi-permeable membrane with pores

that allow water, ions and other small molecules to pass with a Stoke radius below <2.5 nm (**Mohr et al., 2009**), but remains relatively impermeable to large proteins, macromolecular complexes, DNA and RNA, with the exception of specific nuclear transport mechanisms through nuclear pores. Macromolecules produce colloid osmotic pressures, by attracting a shell of water around them (**Mitchison, 2019**; **Vink, 1971**; **Vink, 1974**). For this model on nuclear volume establishment, the relevant colloid osmotic pressures in the cytoplasm ($\pi^{Cy}$) and nucleoplasm ($\pi^N$) are generated by distinct sets of macromolecules that are too large to freely diffuse across the nuclear envelope. These pressures are estimated to be orders of magnitude smaller than turgor pressure (kPa versus MPa in yeast). The apparent absolute numbers of osmotically active molecules in the nucleus and cytoplasm that generate this colloid osmotic pressure are denoted as $N^N$ and $N^{Cy}$, respectively. In addition, there are also non-osmotically active volumes in the cytoplasm and nucleoplasm ($b^{Cy}$, $b^N$), which represent the dry volume taken up by cellular components. The percentage of the dry volume of the nucleus and the cell in isotonic conditions is called the normalized non-osmotic volume (defined as $v_b^N = b^N/V_{iso}^N$ and $v_b^C = b^C/V_{iso}^C$, with $b^C = b^N + b^{Cy}$) such that $V_{iso} - b$ represents the free water within each compartment, and $v_b$ describes the degree of macromolecular crowding.

We postulated that the size of the nucleus is set by a combination of forces that include colloid osmotic pressures of the nucleoplasm and cytoplasm and membrane tensions that restrict expansion of the cell and nuclear membranes ($\sigma$). Membrane tension at the cell surface $\sigma^C$ includes plasma membrane tension as well as other mechanically relevant features such as the cell wall or cortex. Similarly, membrane tension of the nuclear envelope, $\sigma^N$ includes the tension in the inner and outer envelopes and potentially the mechanical properties of the lamina, cytoskeleton, chromatin and factors anchored to the membrane (**Schreiner et al., 2015**). Membrane reservoirs such as eisosomes (**Lemière et al., 2021**) and caveolae (**Sinha et al., 2011**) at the plasma membrane and inner nuclear envelope invaginations and the endoplasmic reticulum for the nuclear envelope (**Fricker et al., 1997**) may reduce membrane tension by allowing for increases in membrane surface area while keeping membrane tension low.

We used established osmotic theory based upon Boyle Van't Hoff's relationship (**Hoff, 1887**) and (**Laplace, 1805**) to analyze the steady state behavior of osmometers in our model. We treated the cell and nucleus as two spherical nested osmometers having respective membrane tensions $\sigma^C$ and $\sigma^N$ and interpreted Van't Hoff's Law in terms of the concentrations of apparent osmotically active particles in the cytoplasm ($C^{Cy}$), nucleoplasm ($C^N$), and extracellular space ($C^{out}$). We described the steady state solutions in which colloid osmotic pressures in the cytoplasm and nucleus are in balance with their respective membrane tension, which results in the coupled equations:

$$\left(C^{Cy} - C^{out}\right) k_B T = 2\sigma^C \frac{4\pi}{3V^C}^{1/3}, \tag{1}$$

$$\left(C^N - C^{Cy}\right) k_B T = 2\sigma^N \frac{4\pi}{3V^N}^{1/3}, \tag{2}$$

where $k_B T$ is the product of Boltzmann's constant and the temperature. Solving this system of equations for the unknown cell volume ($V^C$) and nuclear volume ($V^N$) yields a unique steady-state value for the N/C ratio (Appendix 1). In certain limiting cases, the N/C ratio may be written explicitly in terms of the parameters, as we show later on. However, in general the equations are solved numerically. Note that small molecules that are permeable to the nuclear envelope such as ions do not contribute on their own to the osmotic balance in **Equation 2**.

Using this model, we evaluated what key parameters affect the N/C ratio. To do this, we solved this system of equations (Appendix 1 **Equation A14; A15**) for different sets of parameters to find the resulting N/C ratio. One prediction of this model is that if the normalized non-osmotic volume of the cell equals that of its nucleus ($v_b^C = v_b^N$) then the N/C ratio remains constant under osmotic shifts (**Figure 1B**). Conversely, whenever $v_b^N/v_b^C \neq 1$, the model predicts that the N/C ratio will vary with osmotic shocks (**Figure 1B**). In the case of negligible nuclear tension $\sigma^N \approx 0$ N/m, the N/C ratio remains constant upon osmotic shifts precisely when $v_b^C = v_b^N$ (Appendix 1 **Equation A16**). In **Figure 1C and D** we plotted the effects of varying nuclear membrane tension $\sigma^N$ (from 0 to 2.7 N/m) on the N/C ratio upon osmotic shifts. The results also reveal that the N/C ratio is relatively insensitive to osmotic shocks for small values of $\sigma^N$ independently of $\sigma^C$ ($\sigma^N = 0.5$ mN/m, **Figure 1C and D**, Appendix 1 and 3). **Figure 1E** summarizes these findings on the effects of varying both $\sigma^N/\sigma^C$ and $v_b^C/v_b^N$.

We further considered the limiting case mentioned above of negligible nuclear membrane tension $\sigma^N = 0$ N/m and in which the normalized non-osmotic volumes of the nucleus and cytoplasm are balanced, with $v_b^C = v_b^N$. As explained in Appendix 4, in this case the N/C ratio is set simply by the ratio of the apparent numbers of osmotically active molecules in the nucleoplasm and in the whole cell:

$$N/C_{ratio} = N^{Nucleus}/N^{Cell}. \tag{3}$$

In the sections below, we tested and further developed this osmotic-based model with experiments with fission yeast to measure key parameters and test model predictions.

## The *S. pombe* nucleus behaves as an ideal osmometer

To quantify the osmotic forces that control cell and nuclear size, we experimentally determined the volume responses of fission yeast cells and their nuclei to osmotic shifts in their media. To visualize the cell and nucleus, we imaged fission yeast cells expressing a nuclear membrane marker (Ish1-GFP, *Expósito-Serrano et al., 2020*) and a plasma membrane marker (mCherry-Psy1 (*Kashiwazaki et al., 2011*, *Figure 2A and B*)). We placed live cells in flow chambers and treated them with media containing various concentrations of sorbitol, an osmotic agent (see Methods). Nuclear and cell volumes were measured using a semi-automated 3D segmentation approach (Methods; *Figure 2—figure supplement 1A*). As cells adapt to hyperosmotic shocks by gradually increasing glycerol production to recover their volume (*Chen et al., 2003*), we minimized these adaptation effects by taking measurements acutely upon shocks (<1 min) and by using a *gpd1Δ* mutant background that is delayed in this response (*Hohmann, 2002*; *Minc et al., 2009*, *Figure 2—figure supplement 1A–C*). To analyze volume responses, we used Boyle Van't Hoff (BVH) plots in which the normalized volumes are plotted as a function of normalized inverse concentration in medium (*Figure 2C*). Ideal osmometers are characterized by linear responses following BVH's Law (*Nobel, 1969*), showing that their volume is determined primarily by the osmotic environment with negligible effects of surface tension (*Figure 2C*; dotted line). In contrast, in cases with significant membrane tension, the plots exhibit non-linear responses (*Figure 2C*; green line). Further, the intersection of the BVH plot at the Y-axis provides a measure of the normalized non-osmotic volume ($\nu_b$; *Figure 2C*).

First, we analyzed the effect of osmotic shifts on cellular volume. Hyperosmotic shifts of various sorbitol concentrations caused sizable decrease (up to ~54%) in volume of cells, as previously noted (*Atilgan et al., 2015*; *Knapp et al., 2019*; *Molines et al., 2022*, *Figure 2—figure supplement 1B–C*). The BVH plot showed that the volume responses were non-linear, indicative of a non-ideal osmometer behavior (*Figure 2D*). The relationships were non-linear for both hyper- and hypotonic responses, consistent with the actions of the elastic cell wall that exerts compressive forces on the cell body and resists large expansions of volume (*Atilgan et al., 2015*; *Davì and Minc, 2015*; *Schaber and Klipp, 2008*).

To avoid the effects of the cell wall, we conducted osmotic shift experiments on protoplasts, which are yeast cells in which the cell walls has been enzymatically removed (*Flor-Parra et al., 2014a*; *Lemière et al., 2021*, *Figure 2B*). To maintain viability of protoplasts, sorbitol was added to the medium as osmotic support to substitute for the role of the cell wall and to prevent lysing. We determined the isotonic conditions for these protoplasts to be YE medium supplemented with 0.4 M sorbitol (hereafter called YE +0.4 M), as they had similar cytoplasmic properties as walled cells in YE +0 M sorbitol. At this concentration of sorbitol, an asynchronous population of protoplasts exhibited similar average volumes as those of walled cells in YE +0 M sorbitol, and similar cytoplasmic concentrations as assessed by fluorescence intensity of a cytoplasmic marker E2-mCrimson (Methods, *Figure 2—figure supplement 2A–B*, *Al-Sady et al., 2016*; *Knapp et al., 2019*). For osmotic shift experiments, we prepared protoplasts in this isotonic condition of YE +0.4 M sorbitol ($C_{iso}$), and then shifted them into medium containing a range of sorbitol concentrations below and above the isotonic condition (0.2–1.0 M). These methods allowed for quantitative probing of osmotic effects over a remarkable ~3 fold range of volume; notably, protoplasts were able to swell up to 40% in volume or shrink 40% without bursting.

The BVH plot of protoplast responses showed a linear behavior through this range of sorbitol concentrations (*Figure 2D*), indicative of an ideal osmometer. As the number of osmolytes is directly related to cell volume in osmotic shift experiments for ideal osmometers, this allowed us to estimate *S. pombe* solute concentration at ~$30 \times 10^7$ solutes/$\mu m^3$, which represents an osmolarity of 500±45

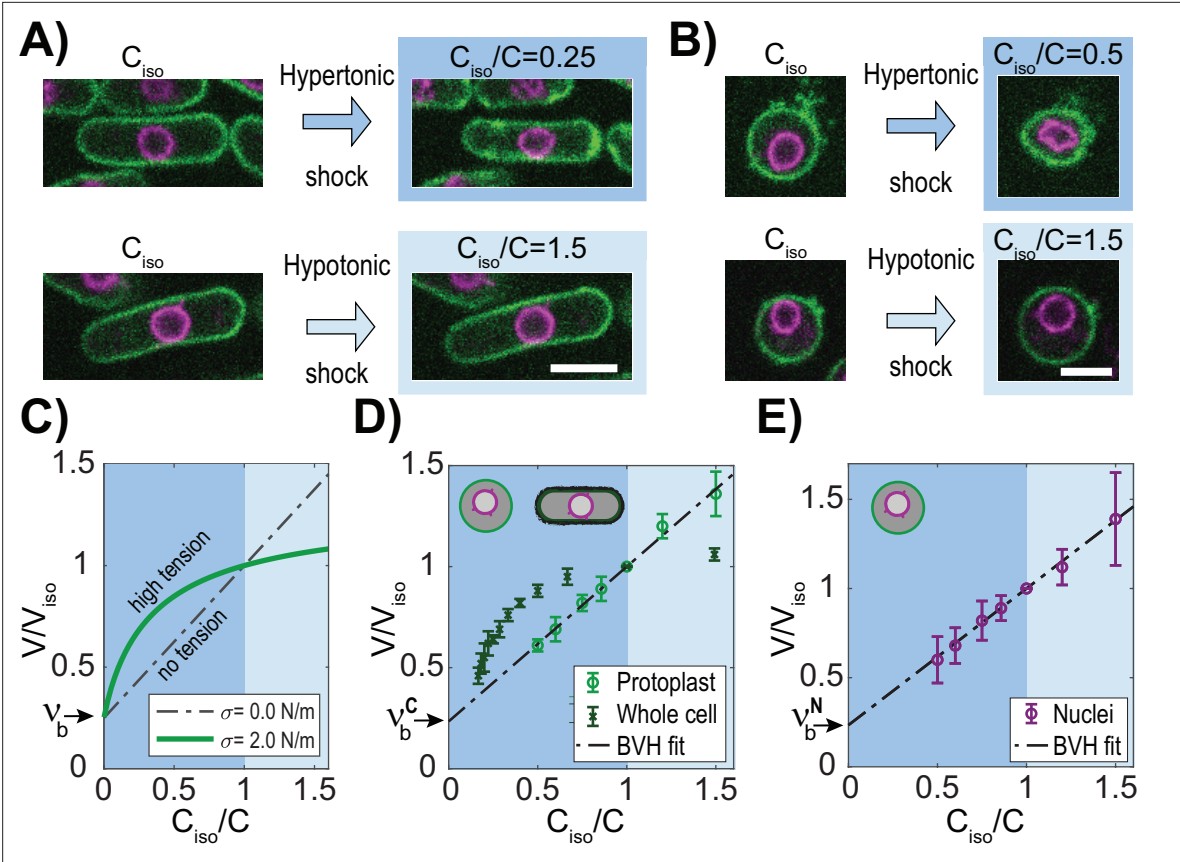

**Figure 2.** The fission yeast nucleus behaves as an ideal osmometer. (**A**) Images of cells expressing a plasma membrane marker mCherry-Psy1 (green) and a nuclear envelope marker Ish1-GFP (purple). Individual cells in isotonic medium ($C_{iso}$) were shifted to hypertonic or hypotonic medium and imaged for 3D volume measurements (Materials and methods). (**B**) Images of individual protoplasts in response to hypertonic and hypotonic shifts. See also *Figure 2—figure supplement 1*. Scale bar = 5 µm. (**C–E**) BVH plots of the effects of osmotic shifts on the volume of the cell and nucleus. (**C**) Theoretical predictions of effects of osmotic concentration in the medium ($C_{iso}/C$) on the volume of a cell or nucleus with zero (black) or large (green) membrane tension ($\sigma$). Dashed line (black) depicts the behavior of an ideal osmometer in which there is no effect of membrane tension. (**D**) Effect of osmotic shifts on the relative volumes ($V/V_{iso}$, mean ± STD) of whole fission yeast cells (N=707, three biological replicates) and protoplasts (N=441, from at least five biological replicates). (**E**) Effect of osmotic shifts on relative nuclear volume ($V/V_{iso}$, mean ± STD) in protoplasts (N=441, from at least five biological replicates). Note that the response of nuclei fits to the predicted behavior of an ideal osmometer.

The online version of this article includes the following source data and figure supplement(s) for figure 2:

**Source data 1.** BVH plots.

**Figure supplement 1.** 3D image analysis methods and use of an osmotic adaptation mutant allow for robust volume measurements.

**Figure supplement 1—source data 1.** N/C ratio comparison.

**Figure supplement 1—source data 2.** WT - *gpd1Δ* background cells volumes over time after a hyper-osmotic shock.

**Figure supplement 1—source data 3.** WT - *gpd1Δ* background cells and nuclei volumes over time after a hyper-osmotic shock.

**Figure supplement 2.** Additional evidence that protoplasts behave as ideal osmometers.

**Figure supplement 2—source data 1.** Cytoplasmic mCrimson concentration.

**Figure supplement 2—source data 2.** Number of osmolytes as a function of cell volume.

mOsmol (Methods, *Figure 2—figure supplement 2C–G*). The BVH plot also showed the normalized non-osmotic volume $\nu_b^C$ to be 25%, similar to what has been previously reported for fission yeast cells (*Atilgan et al., 2015*) and other organisms (*Dill et al., 2011*; *Ellis, 2001*).

Having found that protoplasts behave as ideal osmometers, we then measured how nuclear volume responded to osmotic shocks. In hyperosmotic shifts, nuclei in both whole cells and protoplasts shrank into an abnormal involuted shape, suggesting a loss in volume but not surface area (*Figure 2A and B*), similar to what has been observed in mammalian cells (*Kim et al., 2016*). Strikingly, in hypoosmotic

shifts with protoplasts, nuclei were able to expand in <1 min into a spherical shape with an increase up to 40% in volume and 26% in surface area (*Figure 2B and E*). This large rapid expansion of the nuclear envelope suggested that the nuclear envelope can draw upon membrane stores, potentially from the endoplasmic reticulum (*Fricker et al., 1997*; *Kume et al., 2019*; *Roubinet et al., 2021*). BVH plots showed that the volume of nuclei in protoplasts followed a linear behavior in osmotic shifts over an impressive 3-fold range of volumes (*Figure 2E*). Importantly, this linear response showed that the nucleus behaved as an ideal osmometer. This finding implied that tension of the nuclear envelope was negligible on nuclear size: $\sigma^N \approx 0$ N/m; the nuclear envelope does not exert tension that alters the volume response to osmotic forces, so that nuclear volume is directly responsive to its osmotic environment. The BVH plot also revealed that the normalized non-osmotic volumes in the nucleus $\nu_b^N$ and cytoplasm $\nu_b^C$ (i.e. the dry mass concentration) were similar (25% in nucleus; 25% in cytoplasm) (*Figure 2D and E*).

Thus, these experimental findings show that the protoplast and the nucleus approximate two nested spherical ideal osmometers as described in our theoretical model. As the physical properties of the nucleus are unlikely to change in the short amount of time needed to remove the cell wall, these results imply that the nucleus in whole cells (those with intact cell walls) are also ideal osmometers. Taken together, these findings indicated that fission yeast cells may be represented by the simplest version of the model where $\sigma^N \approx 0$ N/m with matching $\nu_b^C$ and $\nu_b^N$ so that the N/C ratio is determined directly by the ratio of osmotically active molecules in the nucleoplasm to those in the cell.

## The N/C ratio does not depend on the presence of the cell wall

According to the model, the N/C ratio should be independent of the external tension ($\sigma^C$) and the outside concentration ($C^{out}$) (*Figure 1*). To reduce $\sigma^C$ we examined the effects of removing the cell wall. The fission yeast cell wall has an elastic surface modulus of $\sigma^C \sim 10$–$20$ N/m which resists 1.5 MPa of turgor pressure (*Atilgan et al., 2015*; *Minc et al., 2009*). Upon removal of the cell wall, protoplasts are maintained in medium with sorbitol and have a five-orders-of-magnitude decrease in $\sigma^C$, with a membrane tension of $\sim 4.5 \times 10^{-4}$ N/m (*Lemière et al., 2021*). We tracked individual cells during cell wall digestion as they were converted to protoplasts (*Figure 3A*, right panel). There was no significant change in the N/C ratio before and after cell wall removal (*Figure 3A*, left panel). We noted that N/C ratios were slightly elevated in protoplasts in our initial population measurements, but this effect was due to loss of a portion of the cytoplasm trapped in the remaining cell wall during the process of protoplasting (*Figure 3—figure supplement 1A*). Thus, as predicted by our model, the N/C ratio is independent of the outer tension of the system ($\sigma^C$).

## The N/C ratio is maintained under osmotic shifts in protoplasts and whole cells

An important prediction of the osmotic model is that the N/C ratio should not change upon osmotic shifts. To test this prediction, we subjected protoplasts to a range of hypo and hyper shocks and measured nuclear and cellular volumes. We varied sorbitol concentrations from 0.2 to 1.0 M with isotonic conditions defined as 0.4 M sorbitol (*Figure 3—figure supplement 1B and C*). A plot of nuclear versus cellular volumes showed scaling was robustly maintained throughout the range of osmotic conditions (*Figure 3B*, *Figure 3—figure supplement 1D* for separated plots). This was also shown by measurements of the N/C ratios at each sorbitol concentration (*Figure 3C*). Similar experiments in whole cells showed that the distribution of the N/C ratio under osmotic shock (0.1M to 1.0 M) also coincided with the distribution of the same population of whole cells in isotonic condition (*Figure 3D–E*, *Figure 3—figure supplement 1E–G*). Finally, we tested the effect of a hypo-osmotic shock on whole cells. Despite a significant increase of cells and nuclear volumes, the N/C ratio was maintained at 7.3±0.7 (*Figure 3—figure supplement 1H–J*). These results demonstrated that the N/C ratio does not change with the osmotic concentration of the media, confirming the predictions of the model that the sizes of the cell and nucleus are both regulated by osmotic pressures.

## Nanorheology reveals that physical properties of the cytoplasm and nucleoplasm are comparable under osmotic shocks

A key test of the model is to experimentally measure the relevant macromolecular concentrations and intracellular colloid osmotic pressures in both the cytoplasm and nucleoplasm. Recent advances have

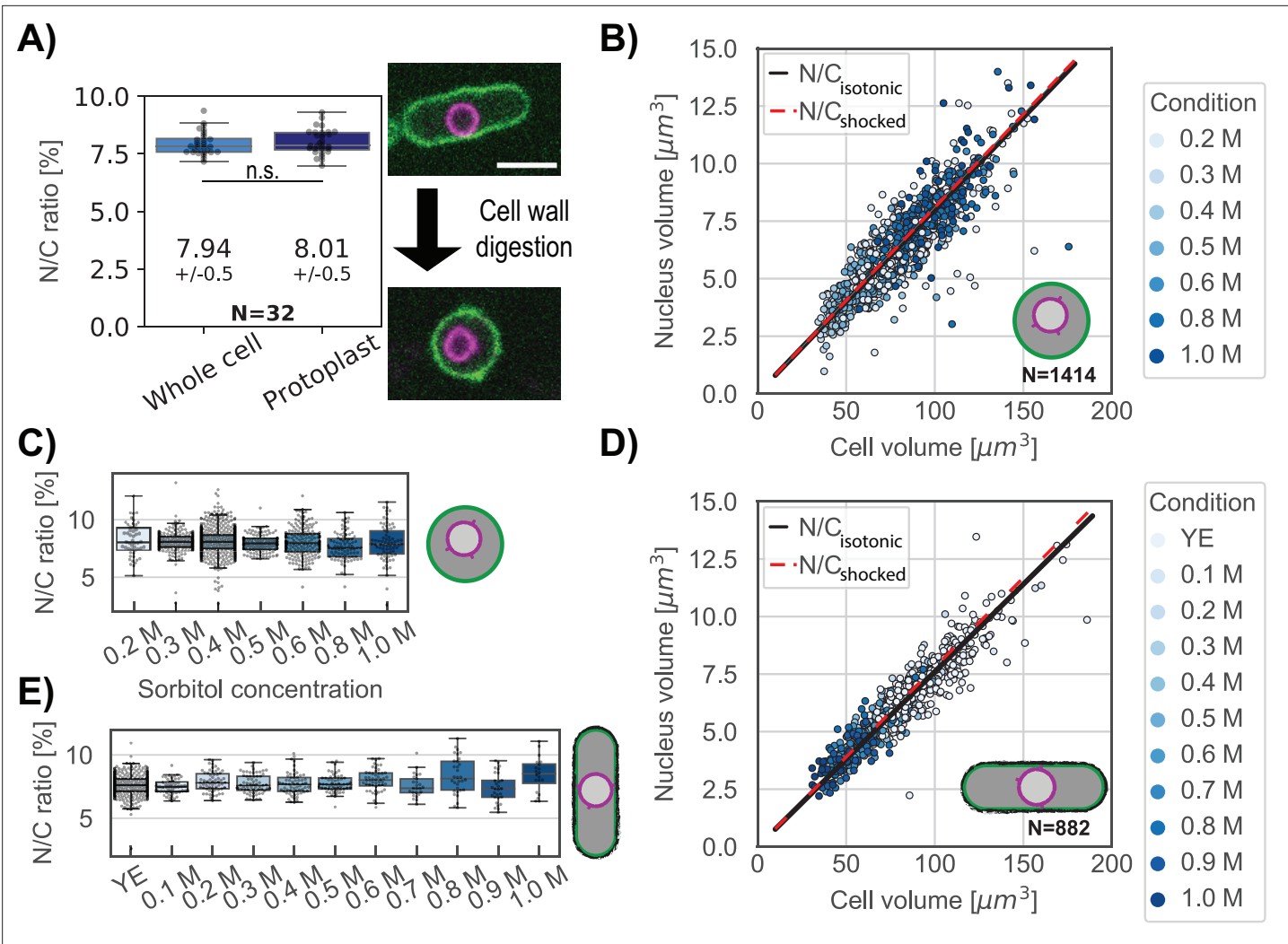

**Figure 3.** The N/C ratio is maintained in osmotic shifts and upon cell-wall removal. (**A**) The N/C ratio of the same cells before and after cell wall digestion (mean ± STD) reveals no statistical differences (paired t test, p=0.36). Right panel, overlay of the plasma membrane (green) and nuclear membrane (purple) of the same cell middle plane before and after cell-wall digestion. For all box and whiskers plots, the horizontal line indicates the median, the box indicates the interquartile range of the data set (IQR) while the whiskers show the rest of the distribution within 1.5*IQR except for points that are defined as outliers. Scale bar = 5 μm. From six biological replicates. (**B**) Scatter plot of cell size and nuclear size for protoplasts under isotonic conditions (YE +0.4 M sorbitol) and immediately following osmotic shocks. Black and dashed red lines, measured N/C ratio of cells under isotonic conditions and osmotically shocked, respectively. From at least two biological replicates. (**C**) Protoplasts with the individual N/C ratio per osmotic condition described in (**B**). (**D**) Same as (**B**) for whole cells in YE under isotonic condition. Black line, measured N/C ratio of cells in isotonic conditions. From two biological replicates. (**E**) Same as (**C**) for whole cells with the individual N/C ratio per osmotic condition described in (**D**). See also *Figure 3—figure supplement 1*.

The online version of this article includes the following source data and figure supplement(s) for figure 3:

**Source data 1.** N/C ratio of the same cells before and after cell wall digestion.

**Figure supplement 1.** Measurements of cellular and nuclear volumes under osmotic shocks in protoplasts and whole cells.

facilitated measurements of these parameters (*Mitchison, 2019*). We used forty nanometer-sized genetically encoded multimeric nanoparticles (GEMs) labeled with mSapphire fluorescent protein as nanorheological probes to quantitatively measure macromolecular crowding through analyses of their diffusive motions (*Delarue et al., 2018*; *Knapp et al., 2019*; *Molines et al., 2022*, *Figure 4A*, green). We used two versions of the GEMs: cytGEM and nucGEM, to measure crowding in the cytoplasm and nucleoplasm, respectively. The nucGEM protein is a version of GEMs that contains a nuclear localization signal (NLS) *Szoradi et al., 2021*; the NLS-GEM monomer is thought to be transported into the nucleus and retained once it assembles into the nucleus with the NLS embedded inside the spherical

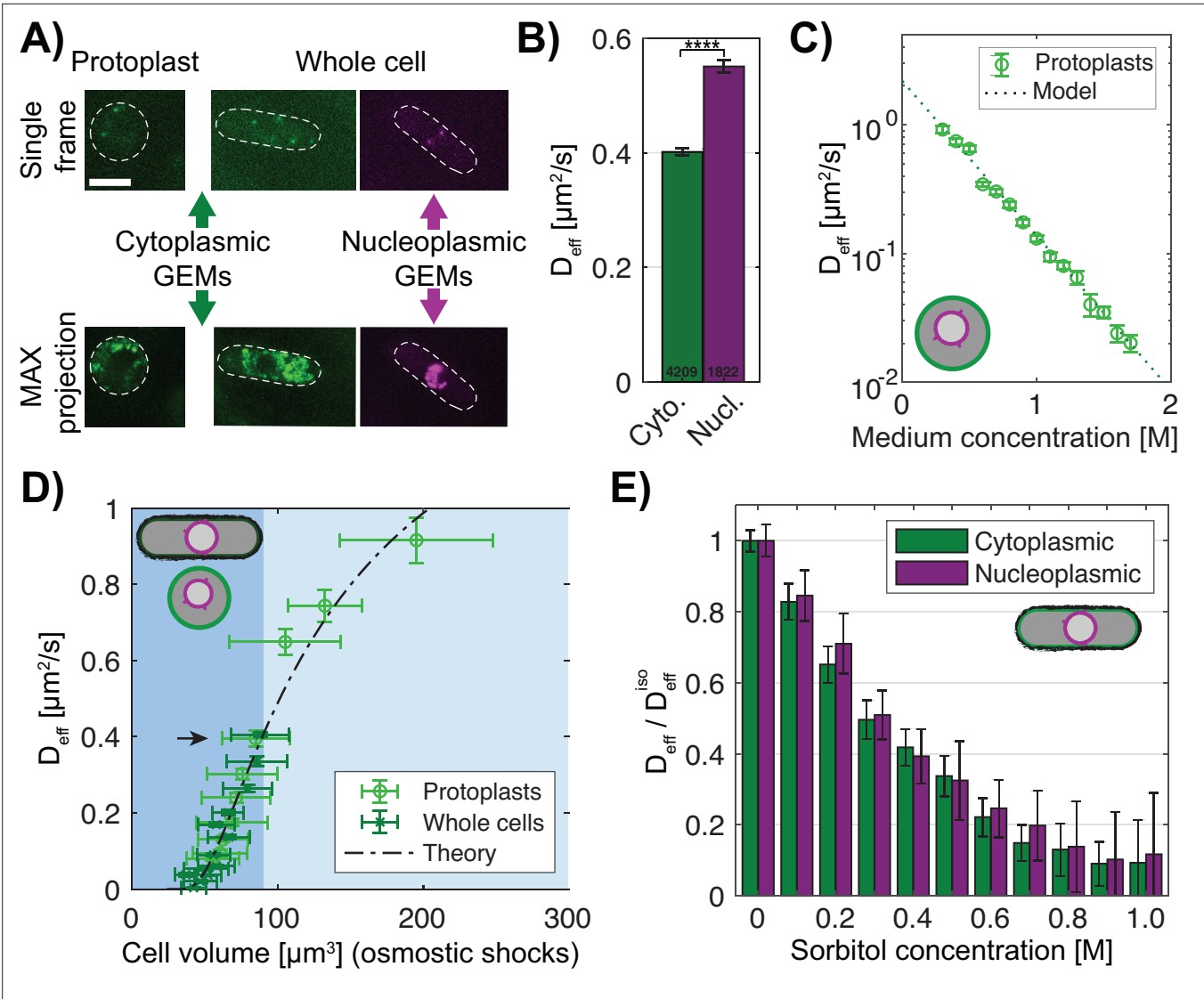

**Figure 4.** Macromolecular crowding is affected similarly in the nucleus and cytoplasm under osmotic shocks. (**A**) Images of protoplasts (left) and whole cells (right) expressing cytoplasmic 40 nm GEMs and nucleoplasmic 40 nm GEMs. Top, single time point image; bottom, maximum projection of 100 frames. Dashed lines, the cell boundary. Scale bar = 5 μm. (**B**) GEMs effective diffusion coefficient (mean ± SEM) is slower in the cytoplasm (green) than in the nucleoplasm (purple) in whole cells in YE medium. Numbers indicate the number of tracks, p-value <0.0001 Mann-Whitney U test. (**C**) Effective diffusion coefficient of cytoplasmic GEMs (mean ± SEM) in protoplasts shifted to various sorbitol concentrations in the medium. Dashed lines, predictions of Phillies' model for diffusion with a power law $\lambda$ =1. $N_{GEMs}$ tracked = 4058, from at least two biological replicates per condition. (**D**) Effective diffusion coefficient of cytoplasmic GEMs (mean ± SEM) plotted against cell volume under hypotonic and hypertonic shock (light blue and blue background respectively). Volumes represent mean distribution of an asynchronous culture ± STD. Dashed line, fit of Phillies' model for self-diffusing trackers in a polymer solution. Black arrow indicates $D_{eff}$ for a population of cells in YE and protoplasts in isotonic condition. Protoplasts: $N_{GEMs}$ = 3355, $N_{Volume}$ = 2216 cells, whole cells: $N_{GEMs}$ = 9849, $N_{Volume}$ = 981 cells. (**E**) Effects of hyperosmotic shifts on the relative effective diffusion coefficients (mean ± SEM) of cytoplasmic and nuclear GEMs; no statistically significant difference was detected (F-test, p-value = 0.90). Cytoplasm $N_{GEMs}$ = 9365,, nuclear $N_{GEMs}$ = 3732, from at least two biological replicates per condition. See also *Figure 4—figure supplement 1*.

The online version of this article includes the following source data and figure supplement(s) for figure 4:

**Source data 1.** Effective diffusion of cytGEMs and nucGEMs.

**Source data 2.** Effective diffusion of cytGEMs in protoplasts in various sorbitol concentrations.

**Source data 3.** Effective diffusion of cytGEMs plotted against cell volume under hypotonic and hypertonic.

**Source data 4.** Normalized effective diffusion of cytGEMs and nucGEMs in various sorbitol concentrations.

**Figure supplement 1.** Comparison of the cytoplasmic and nucleoplasmic GEMs diffusion and anomalous exponent under osmotic shocks.

*Figure 4 continued on next page*

*Figure 4 continued*

**Figure supplement 1—source data 1.** cytGEMs MSD and anomalous diffusion exponent plots in whole cells.

**Figure supplement 1—source data 2.** Effective diffusion of cytGEMs and nucGEMs in whole cells.

**Figure supplement 1—source data 3.** cytGEMs MSD plots and anomalous diffusion exponent in protoplasts.

particle. Cells expressing this NLS-GEMs fusion exhibited motile fluorescent particles in the nucleus (*Figure 4A*). Projections of images over time showed that the nuclear GEMs were excluded from the nucleolus (*Figure 4A*, purple), so that nucGEMs primarily probe the properties of the nucleoplasm outside of the nucleolus.

We compared the behaviors of the GEMs in the cytoplasm and nucleoplasm. Mean square displacement (MSD) curves showed that the cytoplasmic GEMs displayed subdiffusive motion with an anomalous diffusive exponent $\alpha \sim 0.9$ comparable to measurements in HEK293, hPNE cells and *S. cerevisiae* (*Delarue et al., 2018*; *Szoradi et al., 2021*, *Figure 4—figure supplement 1A–B*). Nucleoplasmic GEMs exhibited a stronger subdiffusive behavior with $\alpha \sim 0.8$, suggesting a stronger caging effect compared to the cytoplasm (*Figure 4—figure supplement 1B*). Notably, nuclear GEMs consistently exhibited significantly higher $D_{eff}$ than cytoplasmic GEMs ($D^N_{eff} \sim 0.55$ μm²/s and $D^{Cy}_{eff} \sim 0.40$ μm²/s, *Figure 4B*). These results demonstrated that at the 40 nm size scale, the ability of particles to diffuse is somewhat different in the nucleoplasm versus cytoplasm; these differences may reflect the differences in composition and nanoscale organization between nucleoplasm and cytoplasm. We also assessed cytoplasmic states in protoplasts compared to those in whole cells. Protoplasts in isotonic conditions exhibited similar $D_{eff}$ and α in the cytoplasm, showing that cytoplasmic properties probed by GEMs were not affected by removal of the cell wall (*Figure 4D*, black arrow; *Figure 4—figure supplement 1D and E*).

Next, we determined how $D_{eff}$ of the GEMs relates to macromolecular concentration. Because of the properties of the protoplasts as ideal osmometers, we were able to quantitatively tune macromolecule concentration in the cytoplasm by using osmotic shifts. We found that $D_{eff}$ of the cytoplasmic GEMs in protoplasts exhibited an exponential relationship with medium concentration and hence macromolecular concentration (*Figure 4C*). This relationship could be fit with a Phillies' model (*Masaro and Zhu, 1999*; *Phillies, 1988*) which uses a unique stretched exponential equation to describe a tracer particle's self-diffusive behavior in a wide range of polymer concentrations (Methods, *Figure 4C*, *Figure 4—figure supplement 1F*). The alignment of data from walled cells and protoplasts (*Figure 4D*) showed that this relationship also applied to cytGEMs analyses in walled cells. Hence, these relationships showed that $D_{eff}$ of the cytGEMs can be used to estimate the concentration of macromolecules in the cytoplasm over a large range of concentrations.

We then used the cytoGEMs and nucGEMs to determine how the nucleoplasm compares with the cytoplasm in their responses to osmotic shifts. The proportionate changes of nuclear and cellular volumes (*Figure 3B–D*) predicted that osmotic shifts affect the cytoplasm and nucleoplasm in similar ways. Indeed, even though the absolute values between cytoGEMs and nucGEMs $D_{eff}$ were slightly different, the normalized $D_{eff}$ and α values of cytoGEMS and nucGEMs were similar in cells treated with varying doses of sorbitol (*Figure 4E*, *Figure 4—figure supplement 1B–E*). Together, these findings showed that GEMs can be used to inform on relative changes in the concentration of macromolecules and resultant colloid osmotic pressures within each compartment; for example, both environments showed consistent behavior without evidence for sharp transitions in biophysical properties such as phase transitions. Therefore, these findings demonstrate that the movement of GEMs provides a quantitative approach to assess macromolecular crowding changes within the cytoplasm and nucleoplasm.

## Inhibition of nuclear export causes an increase in the N/C ratio

An important prediction is that changes in the relative numbers of osmotically active macromolecules in the nucleoplasm and cytoplasm would lead to a predictable change in the N/C ratio. It has been previously reported that inhibition of nuclear export leads to an increase in nuclear size, either through treatment with a drug leptomycin B (LMB, an inhibitor of the Crm1 exportin) or through mutants affecting the nuclear transport machinery (*Kudo et al., 1999*; *Kume et al., 2017*; *Neumann and Nurse, 2007*; *Yoshida and Horinouchi, 1999*). LMB causes the redistribution of only a small subset of proteins in *Xenopus* oocytes (*Wühr et al., 2015*). Our model predicted that inhibition of

nuclear export would lead to an increase in macromolecule number in the nucleus relative to that in the cytoplasm. This redistribution would lead to increased osmotic pressure in the nucleus relative to cytoplasm, which would lead to expansion of nuclear volume and/or increased membrane tension. At steady state, in the absence of membrane tension, osmotic pressures and the concentrations of relevant macromolecules would equilibrate at a new larger N/C ratio.

In contrast to previous studies that described effects of LMB after hours of treatment (*Kudo et al., 1998*; *Kume et al., 2019*; *Neumann and Nurse, 2007*; *Nishi et al., 1994*), we examined the acute effects of LMB treatment in a time course, tracking both individual cells (*Figure 5A–E*) and asynchronous populations (*Figure 5—figure supplement 1A–D*). Upon LMB treatment, interphase fission yeast cells continued to grow at a similar rate as untreated cells, but their nucleus grew even faster (*Figure 5A–C*), causing a progressive increase in the N/C ratio from 8% to 9% in an hour (representing a 6% increase of the N/C ratio at 15 min and a 16% increase by 60 min)(*Figure 5D–E*).

We used multiple assays to assess quantitatively the redistribution of macromolecules and osmotic pressure effects in these cells. First, we measured the subcellular localization of ribosomal subunits. Ribosomes and their subunits are major components of biomass and contributors to macromolecular crowding in the cytoplasm (*Delarue et al., 2018*; *Warner, 1999*). Large ribosomal subunit proteins, which are transported into the nucleus to be assembled into pre-60S particles and then exported in a Crm1-dependent manner are known to be inhibited by LMB (*Aitchison and Rout, 2000*; *Ho et al., 2000*). We found that the concentrations of large ribosomal subunits Rpl3001 and Rpl2401 tagged with the GFP (*Knapp et al., 2019*) increased progressively in the nucleus so that by 60 min of LMB treatment, nuclear and cytoplasmic levels were similar (*Figure 5F*, *Figure 5—figure supplement 1E*). In contrast, a small ribosomal subunit protein Rps2-GFP (*Knapp et al., 2019*), which was not expected to be affected by LMB (*Aitchison and Rout, 2000*), showed little accumulation in the nucleus (*Figure 5—figure supplement 1F*). The cytoplasmic intensities of these three ribosomal markers decreased slightly by 45 min (Rpl3001 –19%, Rpl2401 –11% and Rps2-GFP –8%, *Figure 5F*, *Figure 5—figure supplement 1E-F*; right panel), which may be due to redistribution into the nucleus, as well through ribosomal turnover or impaired biogenesis. These examples illustrated how LMB causes a progressive redistribution of a subset of abundant proteins from the cytoplasm into the nucleus.

Second, we quantified concentrations of total protein and RNA by staining fixed cells with the fluorescent dye fluorescein isothiocyanate (FITC) and analyzed their fluorescence intensities (*Knapp et al., 2019*; *Kume et al., 2017*; *Odermatt et al., 2021*). FITC staining intensities in the cytoplasm and nucleoplasm were similar (ratio ~1) in both control cells and those treated with LMB (*Figure 5H*, *Figure 5—figure supplement 1G*). This assay suggested that there was no large redistribution in total protein and RNA. We quantified total protein without the RNA by FITC staining of cells treated with RNAse (*Knapp et al., 2019*; *Odermatt et al., 2021*). In control cells, this staining suggested that total protein concentration was lower in the nucleus than in the cytoplasm (*Figure 5—figure supplement 1H–I*). LMB-treated cells only exhibited a small (~8%) increase at 60 min in the ratio of nuclear to cytoplasmic protein staining compared to control cells (*Figure 5—figure supplement 1I*). This magnitude of protein accumulation in the nucleus was consistent with the observed increase in N/C ratio, as shown by simulations of the effects of redistributing solutes into the nucleus (*Appendix 2—figure 1*).

Third, we used the GEMs-based nanorheology to assess changes in the crowding of macromolecules. $D_{eff}$ of nuclear GEMs showed a small but significant initial decrease at 15 min of LMB treatment, but it subsequently returned to a normal level at 60 min (*Figure 5G*). In contrast, the cytGEMs $D_{eff}$ did not change significantly at 15 min, but increased significantly at 30 and 60 min. These results suggested that there may be a transient small increase in crowding in the nucleus at 15 min, but that crowding levels soon returned to normal; in contrast there was a more impressive progressive dilution of the cytoplasm (equivalent to 4% and 11% dilution at 30 and 60 min, respectively, *Figure 5B*, *Figure 5—figure supplement 1E–F*). LMB may lead to cytoplasmic dilution by inhibiting export of macromolecules such as mRNA out of the nucleus, leading to a decrease in protein synthesis, all while cells continue growing in volume at a normal rate (*Figure 5B*, *Figure 5—figure supplement 1E–F*, *Neurohr et al., 2019*).

Fourth, we determined the effects of LMB on the distribution of non-osmotic volumes $\nu_b$ and nuclear membrane tension. To measure these parameters, we performed osmotic shift experiments on LMB-treated and control protoplasts and analyzed the results using BVH plots (similar to *Figure 2*).

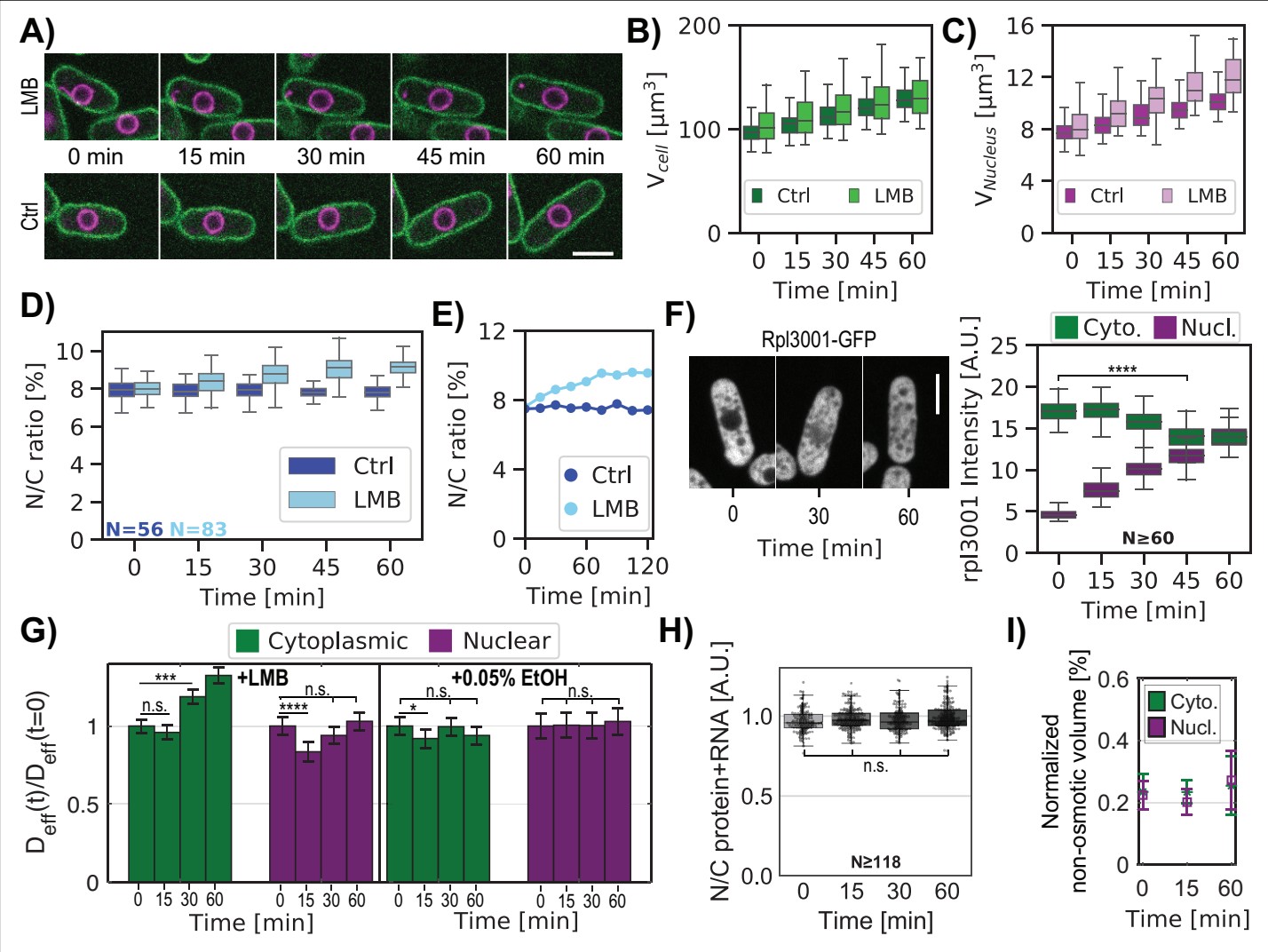

**Figure 5.** Inhibition of nuclear export rapidly leads to an increase in the N/C ratio and changes in crowding. (**A**) Individual cells expressing plasma membrane and nuclear markers were imaged in time upon treatment with LMB or control (Ctrl). Images show a mid-focal plane of plasma membrane (green) and nuclear membrane (purple) treated with LMB (top) or not (Ctrl, bottom) over time (min). (**B**) Cell volumes were measured from 3D images. (**C**) Same as (**B**) for nuclear volume. (**D**) Box and whisker plot of N/C ratio of individual cells treated with LMB (cyan) and control condition (blue) at t=0 min and followed by time lapse microscopy. (**E**) N/C ratio dynamics of representative individual cells extracted from (**D**). (**F**) Cells expressing chromosomally tagged proteins that mark the large ribosomal subunit (Rpl3001-GFP) were treated with LMB and imaged over time. Mid focal plane confocal images and quantitation of their relative fluorescence intensities are displayed. Kruskal-Wallis statistical test was used. (**G**) Cells expressing cytoplasmic or nuclear GEMs were treated with LMB or control (0.05% ethanol) and were imaged for GEMs diffusion over time. Bar graphs show the relative changes in mean effective diffusion coefficients ± SEM for cytoplasmic (green) and nucleoplasmic (purple) GEMs in cells treated with LMB (left panel) and ethanol control (right panel). Statistical differences compared with Mann-Whitney U test. (**A–G**) From at least three biological replicates. (**H**) Cells were stained for total protein and RNA using FITC dye, plots indicate ratios of FITC intensities in nuclear and cytoplasmic regions over time after the addition of LMB. Statistical differences compared with Kruskal-Wallis test (p-value = 0.077), from at two biological replicates. (**I**) Normalized non-osmotic volume over time for cells and their nuclei in protoplasts treated with LMB. Scale bar = 5 μm. See also *Figure 5—figure supplements 1 and 2*.

The online version of this article includes the following source data and figure supplement(s) for figure 5:

**Source data 1.** Cell and nucleus volumes over time treated with LMB.

**Source data 2.** Ribosomal tagged subunit localization in cells treated with LMB.

**Source data 3.** Effective diffusion of cytGEMs and nucGEMs in cells treated with LMB over time.

**Source data 4.** N/C ratio of total protein and RNA over time after the addition of LMB.

**Source data 5.** Normalized non-osmotic volume over time for cells and their nuclei in protoplasts treated with LMB.

**Figure supplement 1.** Effects of LMB on the N/C ratio and ribosomal protein localization.

*Figure 5 continued on next page*

*Figure 5 continued*

**Figure supplement 1—source data 1.** Whole-cell volume and nuclear volume of distinct populations of cells treated with LMB.

**Figure supplement 1—source data 2.** N/C ratio of total protein over time after the addition of LMB.

**Figure supplement 2.** Effects of LMB and osmotic shifts on protoplasts.

**Figure supplement 2—source data 1.** Time course of N/C ratio, cellular volume, and nuclear volume in individual LMB-treated protoplasts.

**Figure supplement 2—source data 2.** BVH plots for protoplasts treated with LMB.

Protoplasts showed a similar increase in the N/C ratio in response to LMB (*Figure 5—figure supplement 2A–D*). The elevated N/C ratio was maintained upon osmotic shocks (*Figure 5—figure supplement 2E–F*); for instance, after 60 min of LMB treatment, protoplasts maintained an elevated N/C ratio of 10% over a range of hypoosmotic and hyperosmotic conditions (*Figure 5—figure supplement 2F*). BVH plots (*Figure 5I*) showed that in cells treated with LMB, the normalized non-osmotic volumes in the nucleus and cytoplasm were maintained at 25%. Thus, as nuclear volume increased, the total amount of non-osmotic volume (i.e. dry mass) increased proportionally so that its ratio remained constant. BVH plots also showed that the nuclei still behaved as ideal osmometers at the 15, 30, and 60 min timepoints (*Figure 5—figure supplement 2G–J*), indicating that membrane tension of the nucleus remained low throughout the time course.

In summary, we quantitated the effects on the nucleoplasm and cytoplasm during the progressive expansion in nuclear volume (16% increase in 60 min) in response to LMB. Various assays showed that LMB treatment not only caused an increase in the number of macromolecules in the nucleus, it also caused a progressive decrease in the number of macromolecules in the cytoplasm. The nucleus, which continued to act as an ideal osmometer, responded to these shifts by equilibrating to a larger size. Adjustments in nuclear volume may therefore maintain normal levels of crowding (GEMs) and density ($v_b$ and FITC staining) in the nucleus.

## Protein synthesis inhibition does not alter the N/C ratio

Another way to globally perturb macromolecular levels is by inhibiting protein synthesis. As LMB treatment disrupted ribosomal biogenesis (*Figure 5F*, *Figure 5—figure supplement 1E–F*), we tested whether inhibition of translation itself would alter the N/C ratio. We analyzed cells treated with 50 mg/mL cycloheximide (CHX, *Polanshek, 1977*, *Figure 6A*). At this relatively low dosage, interphase cells continued to grow in volume but at slower rates (*Figure 6B*). The nuclei also grew at the same slower rate, maintaining the N/C ratio (*Kume et al., 2017*, *Figure 6C, D and E*). This maintenance of the N/C ratio over time was also observed in asynchronous cell populations (*Figure 6—figure supplement 1A–D*). GEMs analyses revealed that $D_{eff}$ of nuclear and cytoplasmic GEMs increased proportionally (*Figure 6F*). Quantification of total protein and RNA on FITC stained cells showed no change in the ratio of nucleoplasmic to cytoplasmic distribution (ratio ~1) (*Figure 6G*, *Figure 6—figure supplement 1E*). However, FITC staining without the RNAs signal indicated a progressive decrease in concentration of total protein in both the nucleus and cytoplasm (*Figure 6H*, *Figure 6—figure supplement 1F–G*), leading to a ~30% decrease in total protein concentration in both compartments after 1 hr of CHX treatment.

These findings demonstrated that the proportionate dilution of macromolecular components in both compartments did not alter the N/C ratio. These experimental results strengthen our model of a nucleus behaving like an ideal osmometer for which a similar decrease in osmotically active particles in both sides of the nuclear envelope leads to a constant N/C ratio.

## N/C ratio homeostasis can be explained by an osmotic model for cell and nuclear growth

The N/C ratio is maintained with little variability, with a coefficient of variation of ~0.1 (*Figure 7A*, WT). The ratio is robustly maintained throughout the course of cell growth during the cell cycle (*Figure 5D*, *Jorgensen et al., 2007*; *Neumann and Nurse, 2007*), indicating that nuclear volume normally grows at the same rate as the volume of the cytoplasm. Like many other cell types, the growth rate of fission yeast cells is largely exponential in character, such that large cells grow faster than smaller ones (*Knapp et al., 2019*; *Pickering et al., 2019*; *Tzur et al., 2009*). One basis for this size dependence is thought to be due to the scaling of active ribosome number in the cytoplasm to cell size. The

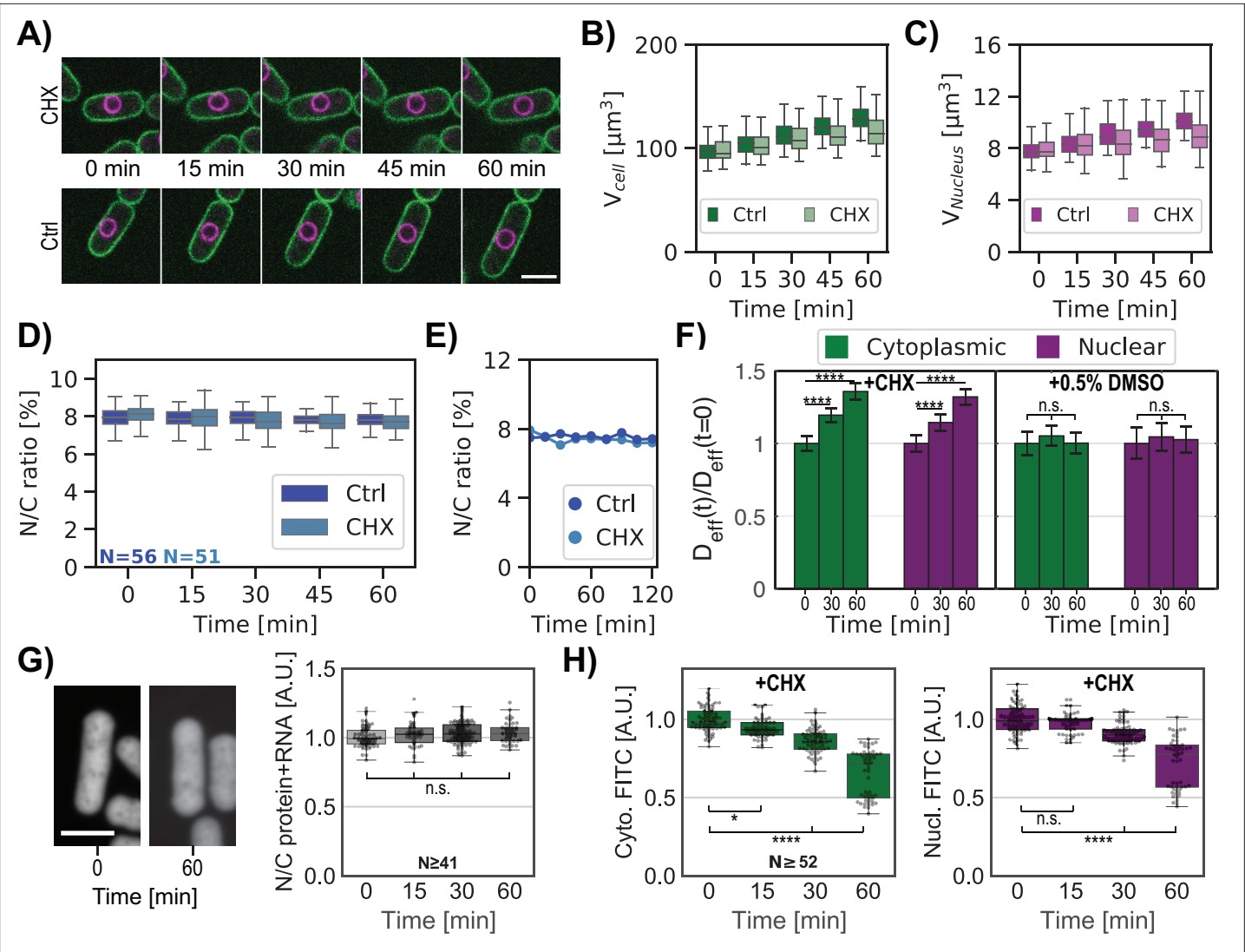

**Figure 6.** Inhibition of protein synthesis is accompanied by a similar decrease in nucleo-cytoplasmic crowding and does not perturb the N/C ratio. (**A**) Overlay of the plasma membrane (green) and nuclear membrane (purple) of whole cells middle plane over time treated with 50 mg/ml cycloheximide (CHX, top) or not (Ctrl, bottom). (**B**) Single whole-cell volume dynamics treated with CHX or not (Ctrl). (**C**) Same as (**B**) for single-nucleus volume dynamics treated with CHX or not (Ctrl). (**D**) Individual whole cells N/C ratio dynamics treated with CHX or not (Ctrl). (**E**) Single whole-cell N/C ratio dynamics extracted from (**D**) for each condition. (**F**) Relative cytoplasmic (green) and nucleoplasmic (purple) GEM effective diffusion dynamics for cells treated with CHX (left panel) or only the drug buffer for control (0.5% dimethyl sulfoxide (DMSO), right panel). Statistical differences compared with Mann-Whitney U test. (**G**) Cells were stained for total protein and RNA using FITC dye. Left, confocal middle plane image of cells before (0 min) or after CHX treatment for 60 min. Right, quantification ratios of FITC intensities in nuclear and cytoplasmic regions over time after the addition of CHX. Kruskal-Wallis statistical test was used, from at two biological replicates. (**H**) Cytoplasmic (green, left) and nucleoplasmic (purple, right) protein signals for the same cells over time under CHX treatment decrease similarly. Kruskal-Wallis statistical test was used. Scale bar = 5 µm. See also ***Figure 6—figure supplement 1***. (A-F &H) From at least three biological replicates.

The online version of this article includes the following source data and figure supplement(s) for figure 6:

**Source data 1.** Whole-cell volume and nuclear volume of distinct populations of cells treated with CHX.

**Source data 2.** Effective diffusion of cytGEMs and nucGEMs in cells treated with CHX over time.

**Source data 3.** N/C ratio of total protein and RNA over time after the addition of CHX.

**Source data 4.** Cytoplasmic and nucleoplasmic protein signals over time under CHX.

**Figure supplement 1.** CHX treatment produces no detectable change in N/C ratio.

**Figure supplement 1—source data 1.** Whole-cell volume and nuclear volume of distinct populations of cells treated with CHX.

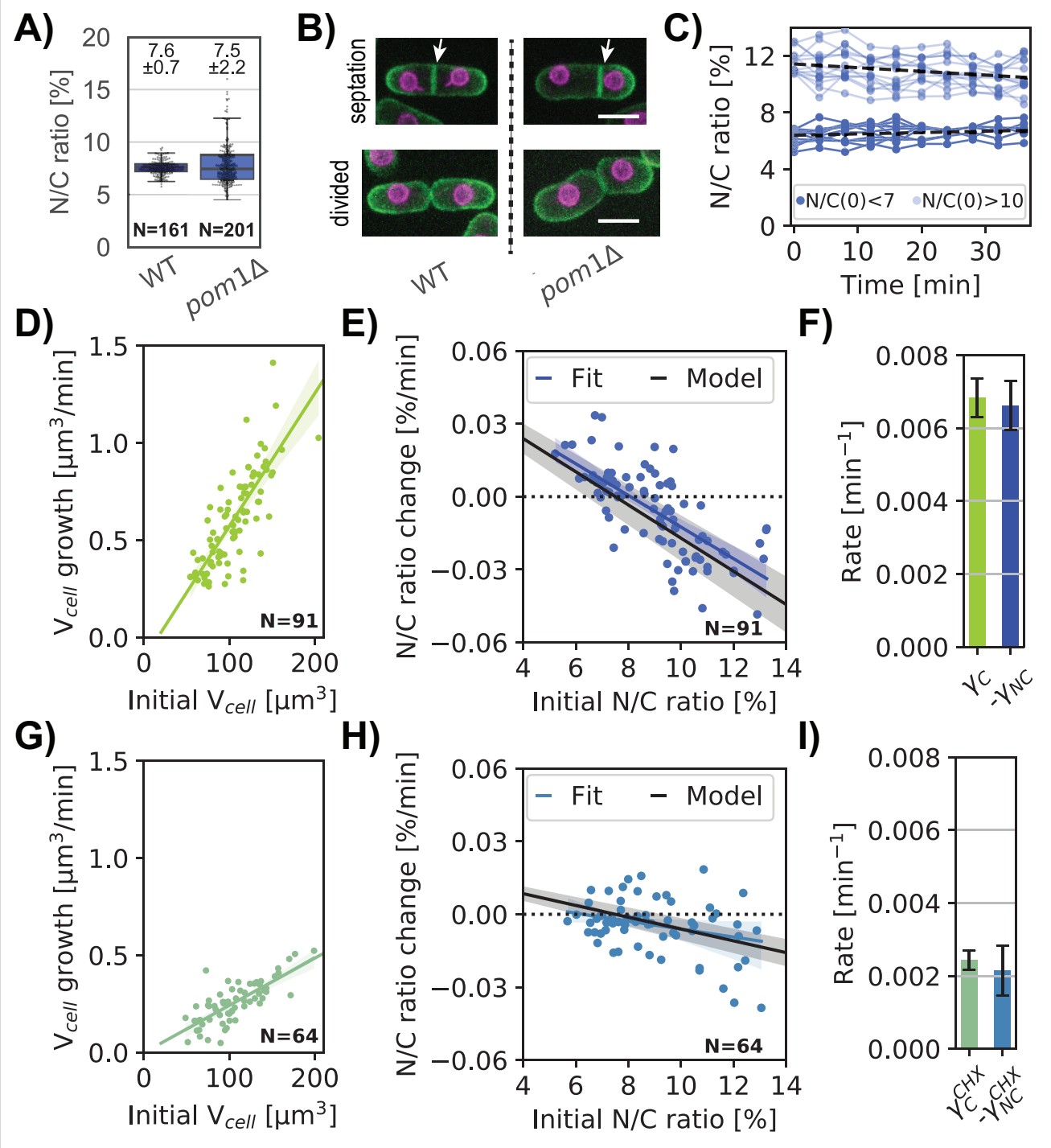

**Figure 7.** Homeostatic correction of aberrant N/C ratios is explained by a passive nuclear growth model. (**A**) Asynchronous WT and *pom1Δ* whole cells N/C ratio (mean ±STD) in growth medium, from 1 biological replicate. (**B**) Z-sum projection overlay of the plasma membrane (green) and nuclear membrane (purple) of representative cells at septation (top) and divided (bottom) for WT (left) and *pom1Δ* (right). White arrow, septum location in the middle of WT cells and decentered for *pom1Δ* cells leading to asymmetric cell division. Scale bar = 5 μm. (**C**) N/C ratio over time for selected cells with low (light blue) or high (dark blue) initial N/C ratio. Dashed lines, linear regression for each cohort of cells. (**D**) Cellular growth rate as a function of a cell's initial volume. Linear regression is shown in green with a slope $\gamma_C$. (**D**) N/C ratio change over time as a function of the initial N/C ratio. Experimental data (blue dots), linear fit (blue line), and predicted passive homeostasis N/C ratio behavior (black line) assuming N/C ratio = 7.5% at equilibrium from (**A**) and cell growth rate $\gamma_C$ from (**C**). See also *Figure 7—figure supplement 1*. (**F**) Comparison of cell growth rate $\gamma_C$ and the N/C ratio correction rate $-\gamma_{NC}$. (**G–I**) Same as D-F but in cells treated with 100 mg/ml CHX to decrease growth rate. (**C–I**) From two biological replicates.

*Figure 7 continued on next page*

*Figure 7 continued*

The online version of this article includes the following source data and figure supplement(s) for figure 7:

**Source data 1.** WT and pom1Δ whole cells N/C ratio, cell, and nuclear volumes.

**Source data 2.** Whole-cell volume, nuclear volume,N/C ratio and cellular growth rate of pom1Δ cells.

**Source data 3.** Whole-cell volume, nuclear volume, N/C ratio and cellular growth rate of pom1Δ cells treated with CHX.

**Figure supplement 1.** Quantitation of cellular and nuclear growth in *pom1Δ* mutant cells.

low variability of the N/C ratio suggests that it may be maintained by a homeostasis mechanism, so that cells with aberrant N/C ratio correct their nuclear size. Indeed, it was recently reported that *S. pombe* cells exhibit homeostasis behavior to maintain nuclear scaling (*Cantwell and Nurse, 2019a*; *Neumann and Nurse, 2007*). We sought to quantify this homeostasis behavior and to test whether the N/C ratio correction could be explained by a passive osmotic model or whether an additional active feedback mechanism needs to be invoked.

We measured homeostasis behavior in *pom1Δ* mutant cells, which display variable N/C ratios because of asymmetric cell division (*Bähler and Pringle, 1998*; *Cantwell and Nurse, 2019a*). Time lapse images showed that these cells exhibited normal mitosis that produced two equally sized nuclei, but often placed their division septum asymmetrically, yielding daughters that were born with either too low or too high N/C ratios (*Figure 7A–B*, *Figure 7—figure supplement 1A–B*). Consistent with this, an asynchronous population of *pom1Δ* cells displayed the same average N/C ratio as wildtype but with a ~threefold larger standard deviation (*Figure 7A*).

We tracked cell and nuclear growth in these cells with abnormal N/C ratio as they grew during interphase (Methods; *Figure 7—figure supplement 1C–E*). Cells born with abnormally large N/C ratios (>10) or abnormally small ratios (<7) gradually corrected their N/C ratio over time (*Figure 7C*, *Figure 7—figure supplement 1E*), consistent with previous findings (*Cantwell and Nurse, 2019a*). Cells exhibited an exponential growth rate of $\gamma_C \approx 0.006$ µm$^3$/min, corresponding to an expected doubling time of ~115 min (*Figure 7D*). Cell growth rate was independent of their N/C ratio (*Figure 7—figure supplement 1F*). In contrast, nuclear growth rate was dependent on the N/C ratio: in cells with high N/C ratios nuclei grew slower than those in cells with normal N/C ratios, while nuclei in cells with low N/C ratios grew faster (*Figure 7—figure supplement 1G*). Cells with near normal N/C ratios showed little change in this ratio over time. These data revealed an inverse relationship between the initial ratio and the rate of correction (*Figure 7E*, blue dots). By fitting the N/C ratio change for a population of *pom1Δ* cells, we quantified the N/C ratio growth rate as a function of the initial N/C ratio (*Figure 7E*, blue line). This homeostasis plot revealed robust N/C ratio homeostasis behavior.

A previous paper proposed a model for N/C ratio homeostasis in which the rate of nuclear growth is function of the N/C ratio, suggestive of an active feedback mechanism ($dV^N/dt = 0.73\left(0.12 - NC\right)$, *Cantwell and Nurse, 2019a*). We tested whether this N/C ratio correction could be instead explained by a passive osmotic model without feedback. We built upon our simple osmotic model to incorporate dynamic growth (See Materials and methods and Appendix 2.2 for detailed derivations). We assumed that volume growth is driven by the rate of biosynthesis of cellular components that scales with the volume of the cytoplasm (*Altenburg et al., 2019*; *Knapp et al., 2019*; *Midtvedt et al., 2019*; *Odermatt et al., 2021*), likely dependent, in part, on the number of active ribosomes in the cytoplasm. The growth of the cytoplasm is driven by the biosynthesis of osmotically-active macromolecules targeted for the cytoplasm. Similarly, the growth of the nucleus is driven by the biosynthesis of macromolecules in the cytoplasm that are transported into the nucleus; assuming that nuclear transport is not limiting, the rate of nuclear growth may thus also scale with the volume of the cytoplasm. The balance of colloid osmotic forces in each compartment determines the cell and nuclear volumes. We also assumed that the rate of synthesis of nuclear components is a fixed percentage of total synthesis rate (e.g. 7.5%). Thus, assuming that the nucleus is an ideal osmometer, the percentage of total synthesis rate of components that end up in the nucleus versus in the cytoplasm is what ultimately sets the N/C ratio at equilibrium.

By assuming exponential cell growth, we can compute the change in N/C ratio over time such that:

$$\frac{dNC}{dt} = \gamma_C \left(f_0 - NC\right) \tag{4}$$

where $f_0$ is a constant that represents the fraction of osmotically active particles transported into the nucleus and $\gamma_C$ is the exponential cellular growth rate. Our measurements gave us access to every parameter in *Equation 4* with no free parameters.

The model predicts that in cells with an altered N/C ratio, the N/C ratio returns to $f_0$ over time, hence $f_0 = 7.5\% \pm 0.7\%$ (*Figure 7A*, *Appendix 2—figure 2*). We used this relationship to determine the homeostasis behavior of the N/C ratio (*Figure 7E*, black line). Our experimental data were an excellent fit for this prediction using the measured parameters with no free parameters ($\gamma_C$ and N/C ratio at equilibrium; (*Figure 7E*; blue versus black lines)).

The model predicts that the rate at which cells correct aberrant N/C ratios $\gamma_{NC}$ is linked to the rate of cell growth ($\gamma_C = -\gamma_{NC}$). Our measurements showed indeed that these rates are similar (*Figure 7F*). It ensues from the model that an alteration in cellular growth rate would cause a proportionate change in the rate of N/C correction. To test this, we treated *pom1* cells with a low dose of CHX that partially inhibits protein synthesis (*Figure 7G*). Under these conditions, the cell growth rate, and the N/C correction rate both decreased about threefold (*Figure 7G,H*) with $\gamma_C^{CHX} = -\gamma_{NC}^{CHX}$ (*Figure 7I*).

These findings show that the continued growth of the cell and nucleus is sufficient to explain the observed correction of the N/C ratio without having to invoke an active mechanism. The correction rates are on the time scales of growth rates, and thus large perturbations in N/C ratio were only partially corrected during a single cell cycle period. Modeling predicted that full correction of significant N/C alterations requires multiple generation times and showed how exponential growth dynamics versus linear growth dynamics affect N/C ratio correction dynamics (*Appendix B -figure 2*).

## Discussion

Here, we provide a quantitative model for nuclear size control based upon osmotic forces. This model, which has zero free parameters, postulates that nuclear size is dictated in part by the numbers of osmotically active molecules in the nucleus and cytoplasm that cannot readily diffuse through the nuclear membranes (*Figure 1*; see Appendix). These molecules, which include large proteins, RNAs, metabolites and other large molecules (>30 kDa), produce colloid osmotic pressure on the nuclear envelope to expand nuclear volume to a predicted size at steady state. Another potential parameter is membrane tension of the nuclear envelope, as determined by its ability to expand under pressure (*Figure 1*). In fission yeast, we determined that the nucleus readily changes in volume under osmotic perturbations, thus behaving as a near-ideal osmometer (*Figure 2E*); this behavior indicates that contributions of membrane tension of the nuclear envelope on nuclear size are negligible (or very small). The N/C ratio is then set as the ratio of nuclear to cytoplasmic solutes *Equation 3*; in the case of fission yeast, the number of these nuclear solutes is predicted to be about 8% of the total in the cell, giving rise to an average N/C ratio of 8%. Therefore, using this system, we confirm and define quantitatively the primary contributions of osmotic pressures to nuclear size control.

This physical model explains why the N/C ratio is so robustly maintained in the vast majority of mutant and physiological conditions (*Cantwell and Nurse, 2019c*). The N/C ratio arises because the cell globally maintains the relative quantities of nuclear to cytoplasmic solutes by protein expression and transport. During cell growth, the nucleus grows at the same rate as the cell because its growth is driven by the synthesis and transport of the nuclear macromolecules that contribute to osmotic pressure. This growth mechanism of the nucleus also explains the homeostasis behavior observed when the N/C ratio is too high or low (*Figure 7*). (*Cantwell and Nurse, 2019a*). The gradual correction of the N/C ratio by growth is reminiscent of 'adder behavior' for cell size homeostasis (*Campos et al., 2014*; *Taheri-Araghi et al., 2015*).

This proposed view suggests that the primary function of nuclear size control is perhaps not to specify a certain size, but to maintain healthy levels of macromolecular crowding in the nucleoplasm (*Ellis, 2001*). Our osmotic perturbations and GEMs measurements showed that the nuclear and cytoplasmic environments, which contain quite different components, nevertheless have similar degrees of mesoscale macromolecular crowding and similar concentrations of non-osmotic volumes. Even after long exposure to LMB, when cells exhibit a significative higher N/C ratio (*Figure 5*, *Figure 5—figure supplement 2A*), the normalized non-osmotic volumes of the nucleus and crowding in the nucleoplasm are similar to those in control cells (*Figure 5I*). The osmotic nature of nuclear size control

thus allows nucleoplasm and cytoplasm to stay in balance through not only synthesis and transport but also by osmotic control of nuclear volume.

Our study provides a critical quantitative confirmation of proposed colloid osmotic pressure effects inside cells (*Harding and Feldherr, 1958*; *Harding and Feldherr, 1959*; *Mitchison, 2019*). Osmotic-based models for nuclear size control have been proposed (*Churney, 1942*; *Kim et al., 2016*), including a recent theoretical report (*Deviri and Safran, 2022*) that evaluates the potential colloid osmotic force contributions of chromatin (1.5 Pa), chromatin counterions (20 Pa), and proteins (8 kPa) to support a similar osmotic model. Our studies here provide quantitative experimental validation in a cell type in which nuclear size happens to be primarily dictated by colloid osmotic forces. The mechanisms by which macromolecules produce colloid osmotic pressure is complex and context dependent (*Mitchison, 2019*). For example, it has been postulated that charged macromolecules and DNA are surrounded by large number of counterions around them and collectively they exert osmotic effects (*Donnan, 1911*). However, under crowded conditions, colloid osmotic pressures generated by contact-interactions between macromolecules may predominate those generated by counterions (*Mitchison, 2019*).

This osmotic-based mechanism is likely the primary factor in nuclear scaling mechanisms in mammalian and other cell types. Osmotic shift experiments in mammalian chondrocytes cells (*Finan et al., 2009*) reveal that the nucleus in these cells is not an ideal osmometer, but is restricted from swelling due to hypoosmotic shocks because of nuclear membrane tension. Indeed, the BVH plot in this paper (Figure 1A of *Finan et al., 2009*) can be fitted with our model assuming $N^N=4.10^{-16}$ mol and a substantial nuclear membrane tension of $\sigma^N = 0.02$ N/m. This additional nuclear membrane tension may be due to nuclear lamina, peri-nuclear actin and/or perinuclear chromatin forces acting on the nuclear envelope (*Edens et al., 2017*; *Newport et al., 1990*; *Schreiner et al., 2015*), which may be absent or reduced in fission yeast . Thus, it is likely that osmotic forces act in concert with other mechanical elements to set nuclear size in these more complex systems. We note that models similar to ours can even account for bacterial nucleoid size scaling (*Gray et al., 2019*), where instead of a nuclear envelope tension term, a partition coefficient establishes the equilibrium concentration of molecules between the cytoplasm and the nucleoid/nucleus. Elucidation of mechanisms rooted in physics promise to give new insights into the range of nuclear shapes and sizes seen during development as well in diseases and aging (*Foraker, 1954*; *Karoutas and Akhtar, 2021*; *Roubinet et al., 2021*; *Zink et al., 2004*). Approaches such as osmotic shifts and nanorheology will allow for future investigation of similar osmotic mechanisms responsible for size control of the nucleus and other organelles.

## Materials and methods

**Key resources table**

| Reagent type (species) or resource | Designation | Source or reference | Identifiers | Additional information |
|---|---|---|---|---|
| Genetic reagent (*Schizosaccharomyces pombe*) | mCherry-Psy1, Ish1-GFP | This manuscript | FC3318 | *h- ade6 <<mCherry-psy1 ish1-GFP:kanMX ura4-D18* |
| Genetic reagent (*S. pombe*) | *gpd1 mutant, mCherry-Psy1, Ish1-GFP* | This manuscript | FC3290 | *h- ade6 <<mCherry-psy1 ish1-GFP:kanMX gpd1::hphMX6 ura4-D18 ade6-* |
| Genetic reagent (*S. pombe*) | *gpd1* mutant | This manuscript | FC3291 | *h- gpd1::hphMX6 ade6-M216 leu1-32 ura4-D18 his3-D1* |
| Genetic reagent (*S. pombe*) | mCherry-Psy1, Cut11-GFP | This manuscript | FC3319 | *h? cut11-GFP:ura4 +ade6:mCherry-psy1 ura4-D18 leu1-32 ade6-M210* |
| Genetic reagent (*S. pombe*) | *gpd1* mutant, CytGEMs | This manuscript | FC3320 | *h- gpd1::hphMX6 pREp41X-Pfv-Sapphire leu1-32 ade6- leu1-32 ura4-D18 his7-366* |
| Genetic reagent (*S. pombe*) | *gpd1* mutant, NucGEMs | This manuscript | FC3321 | *h- gpd1::hphMX6 pREp41X-NLS-Pfv-Sapphire leu1-32 ade6- leu1-32 ura4-D18 his7-366* |
| Genetic reagent (*S. pombe*) | CytGEMs | This manuscript | FC3289 | *h- pREp41X-Pfv-Sapphire ade6-M216 leu1-32 ura4-D18 his3-D1* |

*Continued on next page*

*Continued*

| Reagent type (species) or resource | Designation | Source or reference | Identifiers | Additional information |
|---|---|---|---|---|
| Genetic reagent (*S. pombe*) | NucGEMs | This manuscript | FC3322 | *h- pREp41X-NLS-Pfv-Sapphire leu1-32 ade6-leu1-32 ura4-D18 his7-366* |
| Genetic reagent (*S. pombe*) | *pom1* mutant, mCherry-Psy1, Ish1-GFP | This manuscript | FC3323 | *h- pom1::ura4 ade6 <<mCherry-psy1 ish1-GFP:kanMX* |
| Genetic reagent (*S. pombe*) | Rpl3001-GFP | Chang Lab collection | FC3215 | *h+rpl3001-GFP:kanR leu1-32 ura4-D18 ade6-210* |
| Genetic reagent (*S. pombe*) | Rpl2401-GFP | Chang Lab collection | FC3213 | *h- rpl2401-GFP:kanR leu1-32 ura4-D18 ade6-216* |
| Genetic reagent (*S. pombe*) | Rps2-GFP | Chang Lab collection | FC3209 | *h- rps2-GFP:kanR leu1-32 ura4-D18 ade6-210* |
| Genetic reagent (*S. pombe*) | 1XE2C, GFP-Psy1 | This manuscript | FC3324 | *h+act1p:1XE2C:HygR leu2:GFP-psy1 leu1-ura4-D18 his7-366* |
| Chemical compound/drug | YES 225 Media | Sunrise Science Production | #2011 | |
| Chemical compound/drug | Edinburgh Minimum Media (EMM) | MP Biomedicals | #4110–32 | |
| Chemical compound/drug | Histidine | Sigma-Aldrich | #H8000 | |
| Chemical compound/drug | Uracil | Sigma-Aldrich | #U0750 | |
| Chemical compound/drug | Adenine | Sigma-Aldrich | #A9126 | |
| Chemical compound/drug | Thiamine | Sigma-Aldrich | #T4625 | |
| Chemical compound/drug | Lallzyme | Lallemand | #EL011-2240-15 | |
| Chemical compound/drug | Leptomycin B (LMB) | Alfa Aesar | #87081-35-4 | |
| Chemical compound/drug | Ethanol | Fisher BioReagents | #BP2818-500 | |
| Chemical compound/drug | Dimethyl sulfoxide (DMSO) | Fisher Scientific | #67-68-5 | |
| Chemical compound/drug | Cycloheximide (CHX) | Sigma-Aldrich | #C7698 | |
| Chemical compound/drug | Agarose | Invitrogen | #16500500 | |
| Chemical compound/drug | 4% formaldehyde (methanol-free) | Thermo Scientific | #28,906 | |
| Chemical compound/drug | RNAse | Thermo Scientific | #EN0531 | |
| Chemical compound/drug | Fluorescein isothiocyanate isomer I (FITC) | Sigma | #F7250 | |
| Software, algorithm | µManager v. 1.41 | *Edelstein et al., 2010*; *Edelstein et al., 2014* | | |
| Software, algorithm | Matlab | Mathworks | R2018b | |
| Software, algorithm | Python | *Drake Jr and Van Rossum, 1995* | 5.5.0 | |
| Software, algorithm | Prism | GraphPad | Version 9.3.1 | |
| Software, algorithm | FIJI ImageJ | *Schindelin et al., 2012* | | |
| Other | µ-Slide VI 0.4 channel slide | Ibidi | #80,606 | microfluidic chambers |
| Other | µ-Slide VI 0.5 glass bottom channel slides | Ibidi | #80,607 | microfluidic chambers |

## Yeast strains and media

*Schizosaccharomyces pombe* strains used in this study are listed in Key Resource Table. In general, fission yeast cells were grown in liquid cultures in rich medium YES 225 (#2011, Sunrise Science Production) at 30 °C with shaking. Strains carrying GEM expression vectors were grown in EMM3S – Edinburgh Minimum Media (#4110–32, MP Biomedicals) supplemented with 0.225 g/L of uracil, histidine, and adenine as well as 0.1 μg/mL of thiamine (#U0750, #H8000, #A9126, #T4625, Sigma-Aldrich).

## Microscopy

Cells were imaged on a Ti-Eclipse inverted microscope (Nikon Instruments) with a spinning-disk confocal system (Yokogawa CSU-10) that includes 488 nm and 541 nm laser illumination and emission filters 525±25 nm and 600±25 nm respectively, a 60 X (NA: 1.4) objective, and an EM-CCD camera (Hamamatsu, C9100-13). These components were controlled with μManager v. 1.41 (*Edelstein et al., 2010*; *Edelstein et al., 2014*). Temperature was maintained by a black panel cage incubation system (#748–3040, OkoLab).

For imaging of GEMs, live cells were imaged with a TIRF Diskovery system (Andor) with a Ti-Eclipse inverted microscope stand (Nikon Instruments), 488 nm laser illumination, a 60 X TIRF oil objective (NA:1.49, oil DIC N2) (#MRD01691, Nikon), and an EM-CCD camera (Ixon Ultra 888, Andor), controlled with μManager v. 1.41 (*Edelstein et al., 2010*; *Edelstein et al., 2014*). Temperature was maintained by a black panel cage incubation system (#748–3040, OkoLab).

For most live cell imaging, cells were mounted in μ-Slide VI 0.4 channel slides (#80606, Ibidi – 6 channels slide, channel height 0.4 mm, length 17 mm, and width 3.8 mm, tissue culture treated and sterilized). The μ-Slide channel was coated by pre-incubation with 100 μg/mL of lectin (#L1395, Sigma) for 15 min at room temperature, and then washed with medium. Cells in liquid culture were introduced into the chamber for 3 min and then washed three times with medium to remove non-adhered cells. As certain drugs may adhere to polymer slide material in the conventional chambers, μ-Slide chambers with glass bottoms (#80607, Ibidi – 6 channels slide, channel height 0.54 mm, length 17 mm and width 3.8 mm, D263M Schott glass and sterilized) were used for the drug treatments.

## 3D volume measurements

Nuclear and cell volumes were measured in living fission yeast cells expressing a nuclear membrane marker (Ish1-GFP, *Expósito-Serrano et al., 2020*) and a plasma membrane marker (mCherry-Psy1, *Kashiwazaki et al., 2011*) using a semi-automated 3D segmentation approach. Z stack images (0.5 μm z-slices) that covered the entire cell (for a total of ~20 slices) were obtained using spinning disk confocal microscopy. The 3D volumes were segmented using an ImageJ 3D image segmentation tool LimeSeg (*Machado et al., 2019*; *Schneider et al., 2012*) with these parameters:

For cells: run("Sphere Seg", "d_0=3.0 f_pressure = 0.016 z_scale = 4.5 range_in_d0_units = 3.0 color = 51,153,0 samecell = false show3d=false numberofintegrationstep=-1 realxypixelsize = 0.111"); For nuclei: run("Sphere Seg", "d_0=2.0 f_pressure = 0.016 z_scale = 4.5 range_in_d0_units = 2.0 color = 51,153,0 samecell = false show3d=false numberofintegrationstep=-1 realxypixelsize = 0.111").

After each 3D analysis converged, segmentation results were confirmed using a 2D result. If there was a discrepancy, additional analyses on individual cells were used, with multiple circular regions of interest if necessary. Data were analyzed with Python on Jupiter Notebook 5.5.0. In general, experiments are representative of at least two biological replicates with independent data sets as described in the figure legends.

## Protoplast preparation

*S. pombe* cells were inoculated from fresh agar plates into YES 225 or EMM3S liquid cultures and grown at 30 °C for about 20 hr into exponential phase (OD$_{600}$=0.2–0.3). Ten milliliters of cells were harvested by centrifugation 2 min at 400 rcf, washed two times with SCS buffer (20 mM sodium citrate, 20 mM citric acid , 1 M D-sorbitol, pH = 5.8), resuspended in 1 mL of SCS buffer with 0.1 g/mL Lallzyme (#EL011-2240-15, Lallemand), and incubated with gentle shaking for 10 min at 37 °C in the dark (*Flor-Parra et al., 2014b*). The resulting protoplasts were gently washed three times in YES 225 or EMM3S with 0.4 M D-sorbitol, using centrifugation for 2 min at 400 rcf between washes. After the last wash, 900 μL of supernatant were removed, and the protoplasts in the pellet were gently

resuspended in the remaining ~100 µL of solution. The resultant protoplasts were introduced into a lectin-coated µ-Slide VI 0.4 channel slide for imaging.

### Osmotic shocks

Fission cells or protoplasts were loaded in a lectin-treated µ-Slide VI 0.4 channel slide and maintained at 30 °C. After 5 min of incubation, cells were washed three times with their respective initial buffer (isotonic condition). Cells were imaged first in their initial buffer (isotonic condition). Then, hyper or hypotonic medium was introduced into the channel with three washes. For hypotonic medium YES 225 was diluted with sterile water. The same individual cells were then imaged (within 1 min of the osmotic shift) using the same parameters. To minimize the effect of volume adaptation response to osmotic shock, we assayed cells within 1 min of the osmotic shift and performed most of our experiments using cells in *gpd1∆* mutant background that is defective in this response (*Figure 2—figure supplement 1B and C*, *Hohmann, 2002*; *Minc et al., 2009*).

### Diffusion imaging and analysis of GEMs

For cytoplasmic 40 nm GEMs, Pfv encapsulin-mSapphire was expressed in fission yeast cells carrying the multicopy thiamine-regulated plasmid pREP41X-Pfv-mSapphire (*Delarue et al., 2018*; *Molines et al., 2022*). For nuclear 40 nm GEMs, NLS-Pfv-mSapphire was expressed from a similar pREP41X-NLS-Pfv-mSapphire plasmid (*Szoradi et al., 2021*). The expression of these constructs was under the control of the thiamine repressible *nmt41* promoter (*Maundrell, 1990*). Cells were grown using a protocol that produced appropriate, reproducible expression levels of the GEMs: cells carrying these plasmids were grown from a frozen stock on EMM3S -LEU plates without thiamine for 2–3 days at 30 °C and stored at room temperature for 1–2 days to induce expression. Cells were then inoculated in liquid EMM3S -LEU with 0.1 µg/mL of thiamine (#T4625-25G, Sigma Aldrich) for partial repression of the *nmt41* promoter and grown for one day at 30 °C to exponential phase.

Cells in lectin-treated µ-Slide VI 0.4 channel slides (#80606, Ibidi) were imaged in fields of 250 × 250 pixels or smaller using highly inclined laser beam illumination at 100 Hz for 10 s. GEMs were tracked with the ImageJ Particle Tracker 2D-3D tracking algorithm from MosaicSuite (*Sbalzarini and Koumoutsakos, 2005*) with the following parameters: run("Particle Tracker 2D/3D", "radius = 3 cutoff = 0 per/abs = 0.03 link = 1 displacement = 6 dynamics = Brownian").

The analyses of the GEMs tracks were like those described in *Delarue et al., 2018*, with methods to compute mean square displacement (MSD) using MATLAB (MATLAB_R2018, MathWorks). The effective diffusion coefficient $D_{eff}$ was obtained by fitting the first 10 time points of the MSD curve ($MSD_{truncated}$) to the canonical 2D diffusion law for Brownian motion: $MSD_{truncated}(\tau) = 4 D_{eff} \tau$. In general, experiments are representative of at least 2 biological replicates with independent data sets as described in the figure legends.

### LMB treatment

A stock solution of 0.1 mM LMB (#87081-35-4, Alfa Aesar) in ethanol (#BP2818-500, Fisher BioReagents) was prepared. The final concentration of 25 ng/mL in YES 225 contained 2.3 µL of the stock solution and 5 mL of cell culture. For imaging individual cells over time, exponential phase cells were placed in a µ-Slide VI 0.5 glass bottom channel slide (#80607, Ibidi). Cells where washed three times with a solution of YES 225+25 ng/mL LMB and then imaged. For measurements of a population of cells over time, exponential-phase cells were incubated with the drug at 30 °C with shaking. At each time point, 1 mL of the cell culture was harvested and centrifuged for 2 min at 0.4 rcf. One microliter of the pellet was spread on an 1% agarose (#16500500, Invitrogen) pad (with no drug added), sealed with Valap, and imaged within 5 min.

### Cycloheximide treatment

Cycloheximide (CHX, #C7698, Sigma-Aldrich) stock was prepared at 5 mg/mL in dimethyl sulfoxide (#67-68-5, Fisher Scientific) and stored at –20 °C. CHX was added to a final concentration of 50 µg/mL in *Figure 7* and 100 µg/mL in *Figure 6*.

## FITC staining

Total protein was measured in individual fission yeast cells using FITC staining, similarly as described (*Knapp et al., 2019*; *Odermatt et al., 2021*). One milliliter of cell culture was fixed with 4% formaldehyde (methanol-free solution, #28906, Thermo Scientific, Waltham) for 60 min, washed with phosphate buffered saline (PBS) (#14190, Thermo Scientific,), and stored at 4 °C. One hundred microliters of fixed cells was treated with 0.1 mg/mL RNAse (#EN0531, Thermo Scientific) and incubated in a shaker for 2 hr at 37 °C. Next, cells were washed and re-suspended in PBS and stained with 50 ng/mL FITC (#F7250, Sigma) for 30 min, washed three times, and resuspended in PBS. Cells were mounted on a 1% agarose +PBS pad and imaged in bright field and with 488 nm laser illumination via spinning disk confocal microscopy. The FITC signal was acquired in 300 nm z-step stacks that covered the entire cell volume. For each selected cell, the FITC signal intensities were measured along the long cell axis (averaged over 10 pixels in width) and normalized by cell length. The signal was corrected for background intensity and normalized by the maximum intensity along the line profile within each cell (*Figure 5—figure supplement 1G,H*, *Figure 6—figure supplement 1E,F*). The nuclear and cytoplasmic FITC signals were defined as the sum of the signal from respectively 0.45–0.55 (middle of the cell, for the nucleus) and 0.7–0.8 (for the cytoplasm) along the normalized cell length normalized by the mean value at 0 min.

## N/C ratio homeostasis measurements

*pom1Δ* cells were grown in exponential phase in YES 225, loaded in a µ-Slide VI 0.4 channel slide (#80606, Ibidi), and imaged every 4 min for 40 min at 30 °C. The 3D volumes of each cell and nucleus were measured over time, and interphase growth rates were obtained by extracting the slope of a linear regression to the data over 40 min using a custom-written Python script. Growth rate of mitotic cells were not included in the analysis.

## mCrimson concentration measurements

For *Figure 2—figure supplement 2A and B*, m-Crimson intensity was measured for each cell using the mean fluorescence intensity of a ROI selected in the middle plane of the cell, corrected by the mean fluorescence of the background. The volume of the same cell was measured using the 3D measurement method described above.

## Determination of the intracellular osmolarity in *S. pombe*

For an ideal osmometer, the volume is solely determined by the balance between the outside and the intracellular concentration of osmotically active particles. As we reported that protoplasts behave like ideal osmometers (*Figure 2D*) they can therefore be used to quantify the number of osmolytes ($N^C$) in *S. pombe*. For an ideal osmometer, $N^C$ is directly related to the cell volume ($V^C$):

$$\frac{N^C}{V^C - b^C} = C^{out} \tag{5}$$

We explored the response of protoplast volumes $V^C$ to changes in medium concentration ($C^{out}$) for various osmotic shocks. Protoplasts were prepared in an isotonic solution and shifted in hypo or hyper conditions by the addition or removal of sorbitol in the buffer. The variation of total concentrations $\Delta C^{out} = C^{final} - C^{initial}$ were known. Meanwhile, variations of cells' volume were measured before and after shocks. Since the cell non-osmotic volume $b^C$ does not vary under osmotic shocks, we extracted the only unknown parameter of the equation: each cell's value of $N^C$. We found that $N^C$ is linearly related to the cell volume (*Figure 2—figure supplement 2C–G*), which means that cells keep a constant concentration of osmolytes during the cell cycle. We also confirmed by analyzing various shocks such that $\Delta C^{out}$ spanned from –0.2 to 0.6 M that $N^C$ does not depend on the range of osmotic shocks used to measure it (*Figure 2—figure supplement 2C*). The intracellular osmolyte concentration in *S. pombe* remained constant at a concentration of $\sim 30.10^7$ solutes/µm$^3$.

## Measurement of effective diffusivity

Under acute osmotic perturbations, cell volume changes due to the flow of water, which also affects molecular crowding and the effective GEMs diffusion. We took advantage of these quantitative measurements to assess whether the change in GEMs movements due to osmotic shocks could be

explained with a physical model of diffusion in polymer solution. Phillies' model (*Masaro and Zhu, 1999*; *Phillies, 1988*) uses a unique stretched exponential equation to describe a tracer particle self-diffusive behavior in a wide range of polymer concentrations.

$$D_{eff} = D_0 e^{-\beta C^\lambda} \tag{6}$$

where $D_0$ is the diffusion of the tracer particle in aqueous solution, C is the concentration of polymers, and $\beta$ and $\lambda$ are scaling parameters. $D_0$ can be calculated using the Stokes-Einstein relation for a spherical particle of 40 nm diameter in water. Because protoplasts behave like ideal osmometers (*Figure 2C and D*), their macromolecular intracellular concentration is proportional to the medium concentration $C^{out}$. We took advantage of this behavior to probe the variation of $D_{eff}$ as a function of the medium concentration and found that $\lambda = 1$ fit our data (*Figure 4C*, *Figure 4—figure supplement 1F*). $\lambda$ has been found in in vitro experiments to depend on the molecular weight of the proteins (*Banks and Fradin, 2005*). Interestingly, $\lambda \approx 1$ corresponds to a polymer molecular weight of 43.5 kDa close to the average protein molecular weight for Eukaryotes (~50 kD *Milo and Phillips, 2015*) that fits our in vivo data. Protoplasts behave like ideal osmometers such that we can express the intracellular concentration C in the Phillies' model as a function of the cell volume for each osmotic shock and see whether $D_{eff}$ follows this model for which we now have only one free parameter. The model, assuming a change in intracellular concentration, is in agreement with our experimental values under acute osmotic shifts (*Figure 4C*). We also found that the values for $D_{eff}$ and cell volumes measured on whole cells followed the same model (*Figure 4D*).

## Modeling nuclear growth and N/C ratio homeostasis

We started with a simple model for which nuclear growth was proportional to cell growth while keeping the osmotic behavior of the nucleus: nuclear volume is proportional to the number of osmotically active particles it contains. To determine the cells' growth rate, we imaged cells at 30 °C initially at various stages of the cell cycle every 4 min for 40 min or until division happened. We plotted the change in cells volume as a function their volume at the beginning of the experiment and found a good linear correlation revealing that *S. pombe* growth is exponential with a growth rate $\gamma_C \approx 0.006$ µm³/min (*Figure 7D*) in the same range as previously reported values:

$$\frac{dV^C}{dt} = \gamma_C \ V^C \tag{7}$$

If now we assume that the nuclear growth rate is coupled (by a constant of proportionality) to the cell growth rate (*Figure 7—figure supplement 1H*), then the change in nuclear volume can be written as:

$$\frac{dV^N}{dt} = f_0 \ \gamma_C V^C \tag{8}$$

where $f_0$ is a constant that represents the fraction of osmotically active particles synthesized by the cell that will enter the nucleus. As shown in SM Section S2.2, combining *Equation 7* and *Equation 8* results in *Equation 4* for the rate of change of the N/C ratio.

## Acknowledgements

We are grateful to members of the Chang lab, Sophie Dumont and her lab, Jane Kondev, Ariel Amir and Orna Cohen-Fix for helpful discussions. We thank KC Huang and Gant Luxton for their comments on our manuscript, Catherine Tan for building the pREP41X-Pfv-Sapphire plasmid used in this study, Rafael Daga and Yasushi Hiraoka for providing the ish1-GFP strain, and reviewers for their insightful comments. TGF acknowledges support from National Science Foundation grant DMS-1913093. LJH acknowledges support from American Cancer Society Cornelia T Bailey Foundation, Pershing Square Sohn Cancer Award, Chan Zuckerberg Initiative, NIH R01 GM132447 and R37 CA240765. FC acknowledges support from NIH R01 GM085636, NIH R35 GM141796, NSF/BIO MCB-1638195. The authors JL, PRC and TGF met as participants in the QCBNet Hackathon meeting supported by NSF under MCB-1411898.

## Additional information

### Funding

| Funder | Grant reference number | Author |
|---|---|---|
| National Institutes of Health | R01 GM056836 | Fred Chang |
| National Institutes of Health | R35 GM141796 | Fred Chang |
| National Science Foundation | MCB-1638195 | Fred Chang |
| National Science Foundation | DMS-1913093 | Thomas G Fai |
| American Cancer Society | Cornelia T. Bailey Foundation | Liam J Holt |
| Pershing Square Sohn Cancer Research Alliance | | Liam J Holt |
| National Institutes of Health | R01 GM132447 | Liam J Holt |
| National Institutes of Health | R37 CA240765 | Liam J Holt |
| Chan Zuckerberg Initiative | | Liam J Holt |
| National Science Foundation | MCB Award 2213583 | Fred Chang Thomas G Fai |

The funders had no role in study design, data collection and interpretation, or the decision to submit the work for publication.

### Author contributions

Joël Lemière, Conceptualization, Resources, Data curation, Software, Formal analysis, Funding acquisition, Validation, Investigation, Visualization, Methodology, Writing - original draft, Writing – review and editing; Paula Real-Calderon, Conceptualization, Investigation, Methodology, Writing – review and editing; Liam J Holt, Conceptualization, Resources, Investigation; Thomas G Fai, Conceptualization, Software, Formal analysis, Validation, Investigation, Visualization, Writing - original draft, Project administration, Writing – review and editing; Fred Chang, Conceptualization, Resources, Software, Supervision, Funding acquisition, Validation, Visualization, Methodology, Writing - original draft, Project administration, Writing – review and editing

### Author ORCIDs

Joël Lemière ⓘ http://orcid.org/0000-0002-9017-1959
Paula Real-Calderon ⓘ http://orcid.org/0000-0002-4158-9582
Thomas G Fai ⓘ http://orcid.org/0000-0003-0383-5217
Fred Chang ⓘ http://orcid.org/0000-0002-8907-3286

### Decision letter and Author response

Decision letter https://doi.org/10.7554/eLife.76075.sa1
Author response https://doi.org/10.7554/eLife.76075.sa2

## Additional files

### Supplementary files

• Transparent reporting form

### Data availability

All data generated or analysed during this study are included in the manuscript and supporting file. A source data file has been provided for Figures 2-7 and figure supplements.

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

## Appendix 1

### Mathematical model

We describe a simple theory of nuclear size control based on osmosis, in which the balance of solute particles in the cytoplasm and nucleoplasm determines a unique steady-state nucleus to cell (N/C) ratio.

Our theoretical model is based on the physical mechanism of osmosis. For a perfect osmometer, Van't Hoff's law asserts that the osmotic pressure on a body is determined solely by the relative concentrations of solute particles inside and outside of the body (**Hoff, 1887**). In the context of nuclear size control, we idealize the cell and nucleus as two nested osmometers—with the cell containing the nucleus—and interpret Van't Hoff's law in terms of the concentrations in the cytoplasm, nucleoplasm, and extracellular space.

Cells are densely packed with ions, proteins, and other biomolecules. Without additional constraints, the cell would simply swell osmotically until its internal concentration reached that of the surrounding medium. The elastic cell membrane opposes this osmotic pressure by membrane tension. In this note, we describe how one may calculate steady-state sizes predicted by the balance of forces between membrane tension and osmotic pressure via Van't Hoff's law. The steady-state size depends on the cell's elastic properties along with the solute concentration (or number of osmolytes).

Our model is based on previous investigations in the case of a single compartment, e.g. vesicles consisting of a single lipid bilayer membrane, which were validated by hydrodynamic simulations (**Wu et al., 2015**). In this previous study, the steady-state size could be calculated based on two parameters: the surface tension of the membrane and the number of solute particles.

Following (**Wu et al., 2015**) , we assume that at steady-state each membrane is in a state of mechanical equilibrium. This implies that osmotic pressures $P_{\mathrm{osm}}$ are balanced by mechanical tension $T$ in each membrane. The osmotic pressure $P_{\mathrm{osm}}$ satisfies Van't Hoff's law, which states that $P_{\mathrm{osm}} = c k_B T$ in which $c$ is the concentration of solute particles, $k_B$ is Boltzmann's constant, and $T$ is the temperature. Whereas in **Wu et al., 2015** the assumption of mechanical equilibrium is imposed across a single spherical membrane, here we apply this assumption across both the inner and outer membranes, which results in the coupled equations

$$\left( c^{\mathrm{Cy}} - c^{\mathrm{out}} \right) k_B T = T^C \tag{A1}$$

$$\left( c^{N} - c^{\mathrm{Cy}} \right) k_B T = T^N, \tag{A2}$$

where $T^C$ and $T^N$ are the tensions on the cell and nuclear membranes, respectively, and the external concentration $c^{\mathrm{out}}$ is a parameter that may be tuned experimentally.

In the case of a spherical membrane of radius $R$ subject to surface tension $\sigma$, Young-Laplace's law gives $T = 2\sigma/R$. Assuming the inner and outer membranes have the same surface tension $\sigma$, **Equation A1; Equation A2** become

$$\left( c^{\mathrm{Cy}} - c^{\mathrm{out}} \right) k_B T = \frac{2\sigma^C}{R^C} \tag{A3}$$

$$\left( c^{N} - c^{\mathrm{Cy}} \right) k_B T = \frac{2\sigma^N}{R^N}, \tag{A4}$$

in which $R^C$ and $R^N$ are the radii of the cell and nucleus and $\sigma^C$ and $\sigma^N$ are the respective membrane tensions. Assuming the nucleus and cell are both spherical yields **Equation A1; Equation A2** of the Main Text upon expressing $R^N$ and $R^C$ in terms of their respective volumes.

#### Prescribed concentration

It follows immediately from (**Equation A4**) that to obtain a finite size nuclear membrane the solute concentrations must satisfy $c^N > c^{\mathrm{Cy}}$, in which case the nuclear membrane radius is

$$R^N = \frac{2\sigma^N}{(c^N - c^{\mathrm{Cy}}) k_B T}. \tag{A5}$$

If $c^N - c^{Cy} \leq 0$, both surface tension and osmotic pressure will exert inward forces on the nuclear membrane and its radius will shrink to zero.

Similarly, if the cytoplasm has a prescribed concentration $c^{Cy} > c^{out}$, the resulting steady-state radius $R^C$ will satisfy

$$R^C = \frac{2\sigma^C}{(c^{Cy} - c^{out})k_B T}.$$ (A6)

As above, if $c^{Cy} \leq c^{out}$ both surface tension and osmotic pressure will exert inward forces on the cell membrane and its radius will shrink to zero. In the case of prescribed concentrations $c^N > c^{Cy} > c^{out}$, the resulting ratio of nuclear to cell radii is therefore

$$\frac{R^N}{R^C} = \frac{\sigma^N}{\sigma^C}\frac{c^N - c^{Cy}}{c^{Cy} - c^{out}}.$$ (A7)

## Prescribed solute number

Alternatively, rather than specifying the concentrations, we may instead specify the total numbers of solute particles $N^N$ and $N^C$ contained in the nucleus and cell (so that $N^C - N^N$ is the number contained within the cytoplasm, see *Figure 1(A)* in the Main Text). In this case, Van't Hoff's law may be written in terms of the number of solute particles $N$ and volume $V$. This results in two coupled equations for the membrane radii:

$$\left(\frac{N^N}{V^N} - \frac{N^C - N^N}{V^C - V^N}\right)k_B T = 2\sigma^N\left(\frac{4\pi}{3V^N}\right)^{1/3}$$ (A8)

$$\frac{N^{C,tot} - N^N}{V^C - V^N}k_B T - c^{out}k_B T = 2\sigma^C\left(\frac{4\pi}{3V^C}\right)^{1/3},$$ (A9)

where we have used the relationship $V = (4/3)\pi R^3$ for the volume of a sphere to calculate the radii used in Young-Laplace's law. We remark on the importance of accounting properly for the membrane permeability in the osmotic balance. That is, only those solutes which are impermeable to the membrane may contribute osmotic pressures. In our model, we assume that the cell membrane is impermeable to both macromolecules and ions, i.e. that ion leakage and pumping are not significant on the timescales of interest, whereas the nucleus is impermeable to macromolecules but not ions. This is why $N^{C,tot}$ in (*Equation A9*) must include the number of ions but $N^C$ in (9) only includes the number of macromolecules.

In the case of zero surface tension ($\sigma^N = \sigma^C = 0$), we may simplify the equations above. In this case, (8) simply states that the concentrations are equal in the two compartments, that is

$$\frac{N^N}{V^N} = \frac{N^C - N^N}{V^C - V^N}.$$ (A10)

It follows that the total macromolecule concentration satisfies $N^C/V^C = N^N/V^N$, so that the N/C ratio is given by.

$$\frac{V^N}{V^C} = A,$$ (A11)

where $A = N^N/N^C$ is the ratio of macromolecules in the nucleus and cell. Moreover, from (*Equation A8; Equation A9*), we have.

$$\frac{N^{C,tot}}{V^C} = c^{out},$$ (A12)

so that

$$V^C = \frac{N^{C,tot}}{c^{out}}.$$ (A13)

Assuming the total number of solute molecules are fixed, the above relation (*Equation A13*) predicts a linear relationship between the cell volume and inverse external concentration, with a $y$-intercept at $V^C = 0$.

Experiments including *Figure 2(D)–(E)* in the Main Text agree with the linear correlation but reveal a nonzero $y$-intercept. To account for the nonzero $y$-intercept in the data, we introduce a non-osmotic volume $b$ in both the cell and nucleus, so that the osmotic pressure has the modified form $P_{\text{osm}} = \frac{N}{V-b}k_BT$. This results in the following modifications to *Equation A8; Equation A9*:

$$\left( \frac{N^N}{V^N - b^N} - \frac{N^C - N^N}{V^C - b^C - \left(V^N - b^N\right)} \right) k_BT = 2\sigma^N \left( \frac{4\pi}{3V^N} \right)^{1/3} \tag{A14}$$

$$\frac{N^{C,\text{tot}} - N^N}{V^C - b^C - \left(V^N - b^N\right)} k_BT - c^{\text{out}}k_BT = 2\sigma^C \left( \frac{4\pi}{3V^C} \right)^{1/3}. \tag{A15}$$

Note that $b^C - b^N$ is simply the cytoplasmic non-osmotic volume. The model equations above are similar to Eqns. 5 and 9 in *Deviri and Safran, 2022*, a relevant, independent theoretical study that we learned of while this work was in the late stages of review.

Upon including the non-osmotic volume in this manner, it is straightforward to show that for the case of zero surface tension ($\sigma^N = \sigma^C = 0$) the analogues of (*Equation A11*) and (*Equation A13*) are

$$\frac{V^N}{V^C} = \frac{AN^{C,\text{tot}} + b^N c^{\text{out}}}{N^{C,\text{tot}} + b^C c^{\text{out}}} \tag{A16}$$

$$V^C = \frac{N^{C,\text{tot}}}{c^{\text{out}}} + b^C. \tag{A17}$$

Certain limiting cases of (*Equation A16*) are especially revealing. In the case $b^N/b^C = A$ we have $V^N/V^C = A$ for all $c^{\text{out}}$, recovering the result (*Equation A11*) obtained without non-osmotic volumes. On the other hand, in the limit $c^{\text{out}} \to \infty$, we have $V^N/V^C \to b^N/b^C$ independent of $A$. Although here we have shown this result only in the case of zero nuclear and cell membrane tension, it holds more generally whenever the nuclear membrane tension is zero, as shown in Appendix 4.

Note further that, according to (*Equation A17*), in the case $\sigma^N = \sigma^C = 0$ the cell size $V^C$ is determined only by the total number of solute particles $N^{C,\text{tot}}$ and is independent of the relative ratio $N^N/N^C$. In practice, we find that the cell size stays nearly constant over typical non-zero values of $\sigma^N$ as well, whereas the nuclear volume and N/C ratio increase as the relative nuclear abundance $N^N/N^C$ increases.

In the case of non-zero surface tension, the equations are more difficult to solve analytically. However, it is straightforward to solve *Equation A14; Equation A15* numerically over a range of external osmolarities $c^{\text{out}}$ and to do so we use the command `fsolve` in Matlab.

## Interpretation of solute particles

An important question is whether to include the number of macromolecules, ions, or both in variables $N^N$ and $N^C$. This requires careful consideration of the nature of the compartment and the molecules to which its boundary is permeable.

We therefore consider the relevant osmolytes at both the nuclear and cell membranes. The nuclear pore complex is permeable to ions which can freely cross the nuclear membrane (*Kim et al., 2014*) *Equation A1*. Therefore only macromolecules are able to sustain a concentration gradient, and $N^N$ and $N^C$ in the force balance equation at the nuclear membrane (*Equation A14*) represent the numbers of macromolecules only and do not include ions.

At the cell membrane, on the other hand, ion pumps maintain a concentration gradient. Therefore we must include two types of osmolytes: macromolecules and ions, so that $N^{C,\text{tot}}$ in (*Equation A15*) is given by $N^{C,\text{tot}} = N^C + N^{\text{ion}}$, where $N^C$ and $N^{\text{ion}}$ represent the numbers of macromolecules and ions inside the cell, respectively. In effect, the overall cell size is determined by *both* the macromolecules and the ions, whereas only the number of macromolecules and not the number of ions is important for the N/C ratio.

It is important to note that the number of ions contained in the cell is greater than the number of proteins by approximately two orders of magnitude. This effectively decouples the two equations *Equation A14; Equation A15* , since the cell size is dominated by the number of ions, whereas the N/C ratio is dominated by the numbers of proteins .

**Appendix 1—table 1.** Parameter values and definitions.

| Symbol | Definition | Value | Units | References |
|---|---|---|---|---|
| **Parameters** | | | | |
| $N^{C,\text{tot}}$ | Solute number in cell (including ions) | $2 \times 10^{-14}$ $-8 \times 10^{-14}$ | mol | Estimated from Table 2-3 in *Milo and Phillips, 2015* |
| $N^{C}$ | Solute number in cell (excluding ions) | $2 \times 10^{-16}$ $-8 \times 10^{-16}$ | mol | Estimated from p. 106 in *Milo and Phillips, 2015* |
| $N^{N}$ | Solute number in nucleus (excluding ions) | $2 \times 10^{-17}$ $-6 \times 10^{-17}$ | mol | Fit |
| $\sigma_{\text{intact}}^{C}$ | Cell wall surface tension | 10 $-20$ | N/m | *Minc et al., 2009*, *Atilgan et al., 2015* |
| $\sigma_{\text{proto}}^{C}$ | Cell membrane surface tension | $4.5 \times 10^{-4}$ | N/m | *Lemière et al., 2021* |
| $\sigma^{N}$ | Nuclear membrane surface tension | $4.5 \times 10^{-4}$ | N/m | Estimate |
| $\nu_{\text{iso}}^{C}$ | Cell non-osmotic volume fraction | 0.25 | | Main Text *Figure 2(D)* |
| $\nu_{\text{iso}}^{N}$ | Nucleus non-osmotic volume fraction | 0.25 | | Main Text *Figure 2(E)* |
| | | | | *Figure 3—figure supplement 1* |
| $V_{\text{iso, intact}}^{C}$ | Intact cell isotonic volume | 88 | µm³ | |
| $V_{\text{iso, intact}}^{N}$ | Intact nucleus isotonic volume | 6.7 | µm³ | *Figure 3—figure supplement 1* |
| $V_{\text{iso, proto}}^{C}$ | Protoplast isotonic volume | 81 | µm³ | *Figure 3—figure supplement 1* |
| $V_{\text{iso, proto}}^{N}$ | Protoplast nucleus isotonic volume | 6.6 | µm³ | *Figure 3—figure supplement 1* |
| $c_{\text{iso, intact}}$ | Isotonic sorbitol concentration | 0 | M (mol/L) | *Figure 2—figure supplement 2* |
| $c_{\text{iso, proto}}$ | Isotonic sorbitol concentration | 0.4 | M (mol/L) | *Figure 2—figure supplement 2* |
| **Variables** | | | | |
| $V^{C}$ | Cell volume | | µm³ | |
| $V^{N}$ | Nuclear volume | | µm³ | |
| $c^{\text{out}}$ | External osmolarity | | M (mol/L) | |

# Appendix 2

## Model results

### Response to increasing solute content

Inspired by experiments in which nuclear export is blocked using the drug Leptomycin B (LMB), we next consider the effect of moving solute molecules from the cytosol to the nucleus on the N/C ratio. We start with the following expression for the N/C ratio in the case $\sigma^N = \sigma^C = 0$, in which case:

$$NC = \frac{N^N N^{C,\text{tot}}/N^C + b^N c^{\text{out}}}{N^{C,\text{tot}} + b^C c^{\text{out}}}. \tag{A18}$$

To determine the effect of increasing the number of nuclear solute molecules $N^N$ while holding the total number of solute molecules $N^C$ fixed, we compute

$$\frac{\text{d}(NC)}{\text{d}N^N} = \frac{N^{C,\text{tot}}/N^C}{N^C + b^C c^{\text{out}}}. \tag{A19}$$

According to (**Equation A19**), upon importing a (small) number $\Delta N$ of solute molecules into the nucleus, we expect a corresponding change $\frac{\text{d}(NC)}{\text{d}N^N}\Delta N$ in the N/C ratio. If we enter the estimated values of parameters from **Appendix 1—table 1**, this implies that approximately 10$^{-18}$ mols solute must be imported into the nucleus to generate a 1% change in the N/C ratio. The model predicts that importing a single solute molecule changes the N/C ratio by $2 \times 10^{-7}$%.

Moreover, we may use the measured nuclear and cell sizes during LMB experiments to estimate the solute concentrations and numbers of over time. We find that whereas the concentrations are relatively stable for both protoplasts and intact cells when compared to controls, LMB yields an elevated nuclear solute number upon fitting, consistent with the experimental trend (**Appendix 2—figure 1**).

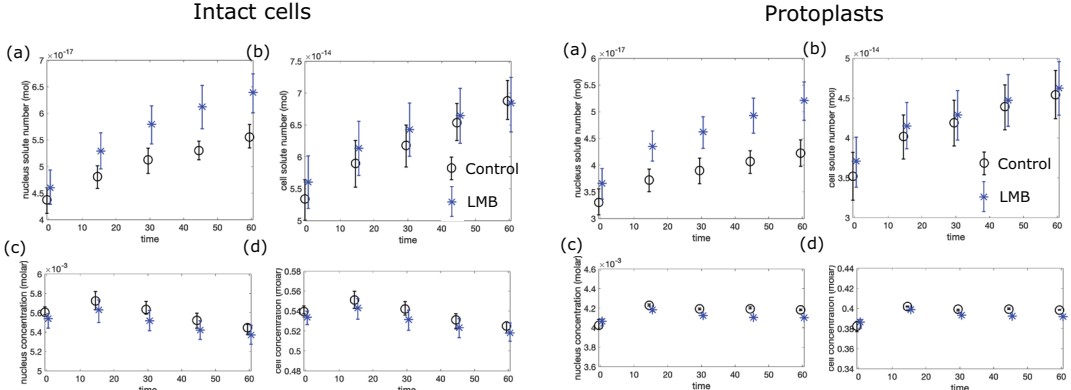

**Appendix 2—figure 1.** Left panel: intact cells, right panel protoplasts. (**a**) Nuclear solute number (mol), (**b**) cell solute number (mol), (**c**) nuclear concentration (**M**), (**d**) cell concentration (**M**). The model predicts the increase in nuclear solute number in LMB experiments upon fitting to the measured cell and nuclear volumes.

### Homeostasis through growth

As discussed in Main Text Methods section "Modeling nuclear growth and N/C ratio homeostasis", the model may be extended to account for cell growth. Upon incorporating cell growth (as discussed next in detail) the model predicts homeostasic behavior in which perturbations in the N/C ratio are corrected over time. Incorporating cell growth in the model leads to two interesting insights. First, exponential growth provides a cell cycle-independent, passive feedback mechanism for correcting aberrations in the N/C ratio. Second, the value of the N/C ratio is determined by the fraction of synthesized proteins that enter the nucleus. These insights provide a plausible genetic explanation for the value of the N/C ratio (the fraction of proteins genetically tagged to enter the nucleus) and raises the question of whether the N/C ratio is a spandrel, i.e. a simple consequence of this fraction rather than a quantity controlled by active feedback.

Next, we derive **Equation A4** in the Main Text. Assuming exponential cell growth yields:

$$\frac{\mathrm{d}V^C}{\mathrm{d}t} = \gamma_C V^C \tag{A20}$$

$$\frac{\mathrm{d}V^N}{\mathrm{d}t} = f_0 \gamma_C V^C. \tag{A21}$$

Applying the quotient rule to $\frac{\mathrm{d}NC}{\mathrm{d}t} = \frac{\mathrm{d}(V^N/V^C)}{\mathrm{d}t}$ yields

$$
\begin{aligned}
\frac{\mathrm{d}NC}{\mathrm{d}t} &= \frac{\dot{V}^N V^C - V^N \dot{V}^C}{(V^C)^2} \\
&= \frac{f_0 \gamma_C (V^C)^2 - \gamma_C V^N V^C}{(V^C)^2} \\
&= \gamma_C \left( f_0 - NC \right),
\end{aligned}
\tag{A22}
$$

where we have used *Equation A20, Equation A21* to obtain the second equality. This indicates a passive feedback mechanism through growth that corrects perturbations in the N/C ratio to the steady-state value of $f_0$.

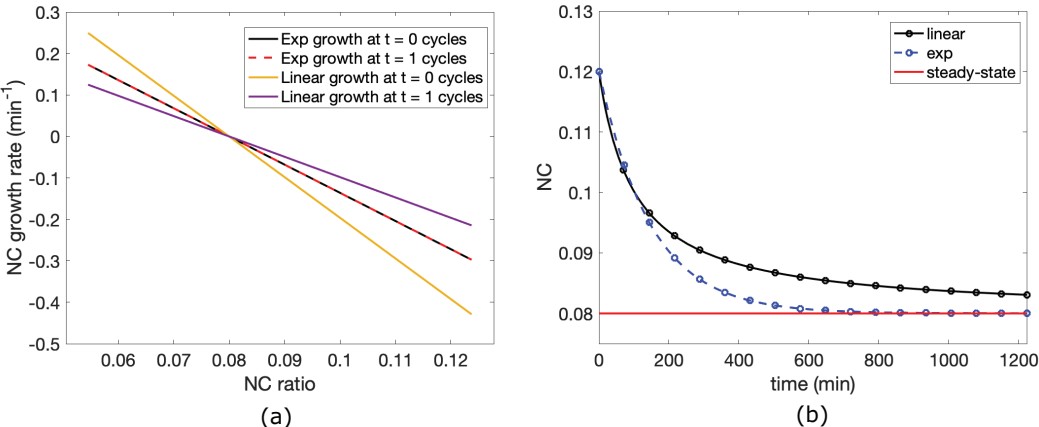

**Appendix 2—figure 2.** Homeostatic behavior predicted by the model in the case of linear and exponential cell growth. (**a**) Exponential growth yields a cell cycle-independent homeostasis, (**b**) Exponential growth leads to a faster recovery of the steady-state N/C ratio in the case of mutants that are unable to divide.

For comparison, we also consider the case of linear growth with a constant growth rate $\alpha$:

$$\frac{\mathrm{d}V^C}{\mathrm{d}t} = \alpha \tag{A23}$$

$$\frac{\mathrm{d}V^N}{\mathrm{d}t} = f_0 \alpha. \tag{A24}$$

In this case, following the same steps as above yields

$$
\begin{aligned}
\frac{\mathrm{d}NC}{\mathrm{d}t} &= \frac{\dot{V}^N V^C - V^N \dot{V}^C}{(V^C)^2} \\
&= \frac{f_0 \alpha V^C - \alpha V^N}{(V^C)^2} \\
&= \frac{\alpha}{V^C} \left( f_0 - NC \right).
\end{aligned}
\tag{A25}
$$

Linear growth also yields a passive homeostasis in the N/C ratio, but with the notable difference that in this case the correction rate is cell-size dependent. This is because of the factor $V^C$ in the denominator of (*Equation A25*). If one calibrates the linear growth model so that the cell doubling time is equal to that of exponential growth, this cell-size dependence manifests as a faster correction rate early on in the cell cycle, and a slower correction rate later on in the cell cycle (*Appendix 2—figure 2(a)*).

One consequence of this comparison between homeostatic behavior for linear and exponential growth is that, in mutants in which the cell and nucleus are unable to divide, exponential growth leads to a much more rapid return to homeostasis than linear growth (*Appendix 2—figure 2(b)*).

## Appendix 3

### Nondimensionalization

We next introduce dimensionless variables in order to consider the limiting cases of large and small surface tension on the inner and outer membranes. In what follows, we modify the notation to make the presentation more general.

Let $\widetilde{V} = N^{C,\text{tot}}/c^{\text{out}}$ denote the characteristic cell volume, $A = N^N/N^C$ be the ratio of solute molecules as above, and $\epsilon^N = 2\left(\frac{4\pi}{3}\right)^{1/3}\left(\frac{1}{N^{C,\text{tot}}(c^{\text{out}})^2}\right)^{1/3}\left(\frac{\sigma^N}{k_B T}\right)$ and $\epsilon^C = 2\left(\frac{4\pi}{3}\right)^{1/3}\left(\frac{1}{N^{C,\text{tot}}(c^{\text{out}})^2}\right)^{1/3}\left(\frac{\sigma^C}{k_B T}\right)$ be the nondimensional nuclear and cell membrane tension, respectively. Note that the $\epsilon$'s are indeed dimensionless groups, since surface tension having units of $J/m^2$ implies that $\sigma/k_B T$ has units of $1/m^2$, whereas $(1/c^{\text{out}})^{2/3}$ has units of $m^2$.

In terms of the dimensionless volumes $v^N = V^N/\widetilde{V}$ and $v^C = V^C/\widetilde{V}$, non-osmotic volumes $\nu_b^N = b^N/\widetilde{V}$ and $\nu_b^C = b^C/\widetilde{V}$, and ratio of macromolecules to osmolytes $\theta = N^C/N^{C,\text{tot}}$, *Equation A14; Equation A15* may be rewritten as

$$\frac{A}{v^N - \nu_b^N} - \frac{1-A}{v^C - \nu_b^C - \left(v^N - \nu_b^N\right)} = \frac{\epsilon^N}{\theta}\left(\frac{1}{v^N}\right)^{1/3} \tag{A26}$$

$$\frac{1-\theta A}{v^C - \nu_b^C - \left(v^N - \nu_b^N\right)} - 1 = \epsilon^C\left(\frac{1}{v^C}\right)^{1/3}, \tag{A27}$$

To study the experimentally-relevant case in yeast, we set $\epsilon^N = 0$. Therefore

$$\frac{A}{v^N - \nu_b^N} = \frac{1-A}{v^C - \nu_b^C - \left(v^N - \nu_b^N\right)} \equiv c. \tag{A28}$$

Of course, the overall concentration $1/\left(v^C - \nu_b^C\right)$ must also be equal to $c$ as well. This can be shown as follows:

$$\frac{1}{v^C - \nu_b^C} = \frac{c(v^N - \nu_b^N) + c\left(v^C - \nu_b^C - \left(v^N - \nu_b^N\right)\right)}{v^C - \nu_b^C} = c. \tag{A29}$$

Therefore, in the case of interest $\epsilon^N \approx 0$ we may rewrite (27) as

$$\frac{1}{v^C - \nu_b^C} - 1 = \epsilon^C\left(\frac{1}{v^C}\right)^{1/3} \tag{A30}$$

to obtain a single equation for the single unknown $v^C$. Next, we consider the asymptotics of solutions to this equation in the limits of large and small $\epsilon^C$.

### Limit of $\epsilon^C \to 0$

First, we define an auxiliary variable $x = (v^C)^{1/3}$ to transform (*Equation A30*) into the following equation in $x$:

$$\frac{1}{x^3 - \nu_b^C} - 1 = \frac{\epsilon^C}{x}. \tag{A31}$$

Multiplying through by the common denominator $x\left(x^3 - \nu_b^C\right)$, (*Equation A31*) may be rewritten as a quartic polynomial:

$$x^4 + \epsilon^C x^3 - \left(1 + \nu_b^C\right)x - \epsilon^C \nu_b^C = 0. \tag{A32}$$

that is $f(x) = 0$ with $f(x) = x^4 + \epsilon^C x^3 - \left(1 + \nu_b^C\right)x - \epsilon^C \nu_b^C$. Because it a fourth-order polynomial, $f$ has four roots, some of which may be complex-valued or negative. However, we will show that there is a unique physical solution corresponding to a positive value of $x$.

Next, we apply the method of dominant balance to approximate the roots. It is straightforward to show that there are two valid balances in the limit $\epsilon^C \to 0$. First, there is the one obtained by dropping the term involving the small parameter, that is

$$x^4 - \left(1 + \nu_b^C\right) x = 0, \tag{A33}$$

which has a single real root at $x_1 = \sqrt[3]{1 + \nu_b^C}$ in addition to two complex-valued roots. Another valid dominant balance is obtained by dropping the two higher-order terms:

$$- \left(1 + \nu_b^C\right) x - \epsilon^C \nu_b^C = 0, \tag{A34}$$

which yields the (non-physical) negative solution $x_2 = -\epsilon^C \nu_b^C / (1 + \nu_b^C)$. In practice, we therefore expect solutions to (*Equation A32*) to approach the positive $x_1$ as the outer surface tension becomes smaller and smaller.

### Limit of $\epsilon^C \to \infty$
In this limit, there are two possible balances, each of which retain the large parameter. It is straightforward to show that these are

$$x^4 + \epsilon^C x^3 = 0, \tag{A35}$$

which yields the solution $x_3 = -\epsilon^C$ and

$$\epsilon^C x^3 - \epsilon^C \nu_b^C = 0, \tag{A36}$$

which yields the solution $x_4 = \sqrt[3]{\nu_b^C}$. In *Appendix 3—figure 1*, we fix $\nu_b^C = 0.25$ and illustrate the convergence of the roots of (*Equation A32*) to the asymptotic limits given by $x_1$ and $x_2$ in the small-$\epsilon^C$ regime and to $x_3$ and $x_4$ in the large-$\epsilon^C$ regime.

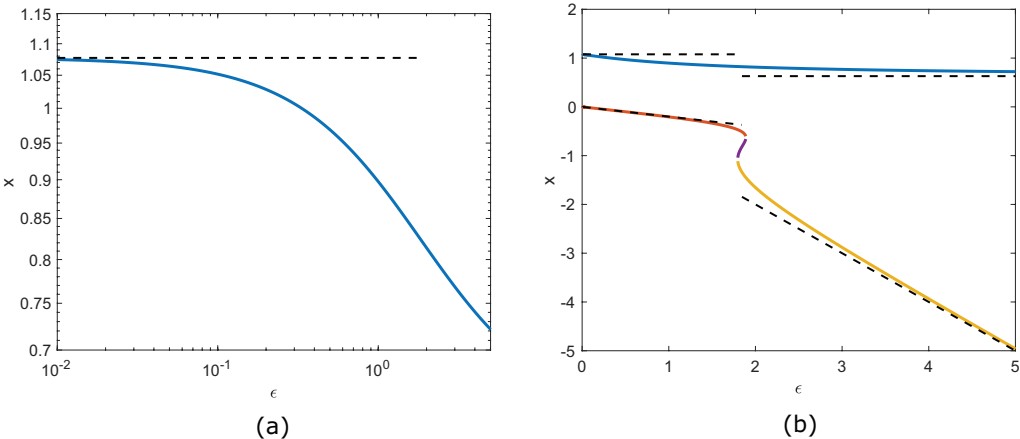

**Appendix 3—figure 1.** Limiting behaviors. (**a**) Convergence to $x_1$ in the small $\epsilon^C$ limit, (**b**) Full range of $\epsilon^C$ ($\nu_b^C = 0.25$).

### Transition between limits
As shown in *Appendix 3—figure 1*, there is a critical interval in which (*Equation A32*) possesses four real roots. This critical interval is located between the small and large-$\epsilon^C$ regimes, and therefore we interpret the midpoint of this interval $\epsilon_*^C \approx 1.84$ to be the transition between the external osmolarity and surface tension-dominated regimes.

To determine the corresponding critical surface tension $\sigma_*^C$, we use the definition of $\epsilon^C$ together with the values found in *Appendix 1—table 1* to estimate $\sigma_*^C \approx 3.8$ N/m. *Appendix 3—figure 2* shows (in red) the Boyle-Van't Hoff plot associated with the critical surface tension $\sigma_*^C$, which as expected occurs between the linear (external osmolarity dominated) and highly nonlinear (surface tension dominated) regimes. The fact that $\sigma_{\text{proto}}^C < \sigma_*^C$ whereas $\sigma_{\text{intact}}^C > \sigma_*^C$ is consistent with the experimental observation that the protoplasts follow a Boyle Van't-Hoff law scaling behavior whereas the intact cell does not.

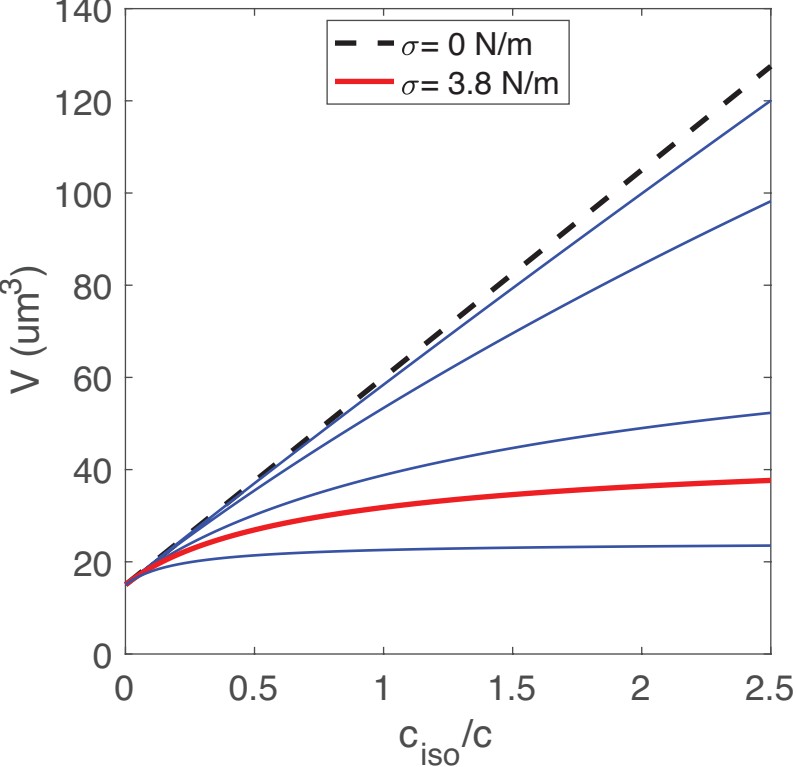

**Appendix 3—figure 2.** Boyle Van't Hoff plots for different values of the surface tension $\sigma^C$. The curve in red corresponds to the critical tension $\sigma^C_* \approx 3.8$ N/m, whereas the dotted line corresponds to $\sigma^C = 0$.

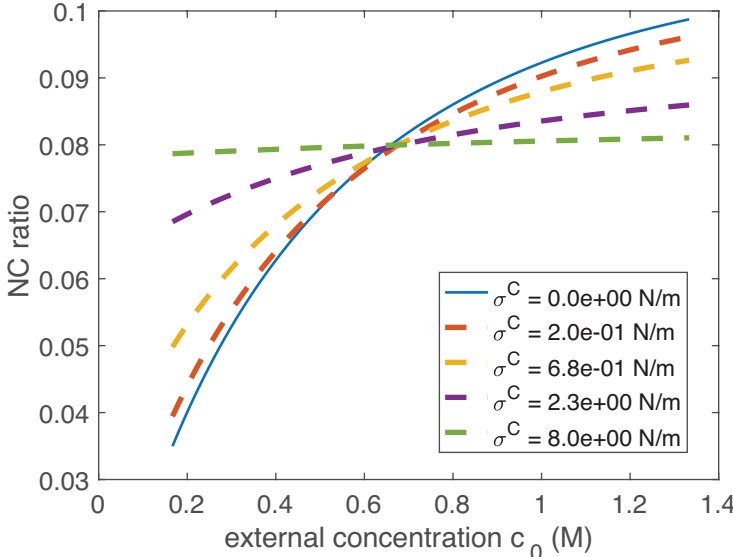

**Appendix 3—figure 3.** Limiting behavior upon varying the outer tension $\sigma^C$ in the limit of large inner tension $\sigma^N = \sigma^C_{\text{intact}}$.

In addition to the limiting case above in which the inner membrane tension $\sigma^N$ is negligible, we also consider the solutions to **Equation A14; Equation A15** in the inverted case of large $\sigma^N$ and negligible outer tension $\sigma^C$. This situation is likely to be the relevant one for mammalian cells, which

do not have a cell wall to provide rigidity to the outer membrane but contain lamins in the nuclear membrane that provide additional structural rigidity compared to yeast.

In *Appendix 3—figure 3*, we plot the N/C ratio in this case $\sigma^N = \sigma^C_{\text{intact}}$ for several choices of the outer tension $\sigma^C$. Unlike the previous limit in which the inner tension is small, in this inverted case the N/C ratio can be seen to vary considerably in response to osmotic shocks. Based on these model predictions we therefore expect that, unlike the nearly constant N/C ratio maintained by force balance in yeast, maintaining a stable N/C ratio in mammalian cells in response to osmotic shocks would require additional control mechanisms over solute concentrations and/or plasma volumes.

## Appendix 4

## Limiting case: balanced non-osmotic volumes at zero tension

Here we show that, if the nuclear membrane satisfies $\sigma^N = 0$ and the non-osmotic volumes $b^N$ and $b^C$ satisfy $b^N/b^C = V_{iso}^N/V_{iso}^C$, i.e. the non-osmotic volumes are in proportion to the nuclear and cell volumes at some reference concentration, then the N/C ratio satisfies

$$V^N/V^C = A \tag{A37}$$

independent of the external concentration $c^{out}$.

We proceed with the derivation of (*Equation A37*). In the tension-free case, the force balance condition analogous to (*Equation A10*) in the presence of non-osmotic volumes is.

$$\frac{N^N}{V^N - b^N} = \frac{N^C}{V^C - b^C}, \tag{A38}$$

where as above $N^N$ and $N^C$ are the number of solute molecules in the nucleus and cell, respectively. Therefore.

$$\frac{V^N - b^N}{V^C - b^C} = \frac{N^N}{N^C}. \tag{A39}$$

This equation is true regardless of the external concentration and so far we have not invoked any hypothesis on the $b_i$. In particular, at the isotonic concentration at which $V^N = V_{iso}^N$ and $V^C = V_{iso}^C$, we have.

$$\frac{V_{iso}^N - b^N}{V_{iso}^C - b^C} = \frac{N^N}{N^C}, \tag{A40}$$

or equivalently.

$$\frac{V_{iso}^N (1 - \nu^N)}{V_{iso}^C (1 - \nu^C)} = \frac{N^N}{N^C}, \tag{A41}$$

where $\nu^N = b^N/V_{iso}^N$ and $\nu^C = b^C/V_{iso}^C$ are dimensionless non-osmotic volumes.

It follows that the N/C ratio at the isotonic concentration, denoted by $NC_{iso}$, satisfies.

$$NC_{iso} = \frac{N^N}{N^C} \frac{1 - \nu^N}{1 - \nu^C}. \tag{A42}$$

Next, we find a formula for the N/C ratio that holds for arbitrary external concentrations. We recall (Equation A39), which asserts that as a consequence of force-balance.

$$\frac{V^N - b^N}{V^C - b^C} = \frac{N^N}{N^C}. \tag{A43}$$

Multiplying through both sides by $V^C - b^C$ yields.

$$V^N = \frac{N^N}{N^C} \left( V^C - b^C \right) + b^N, \tag{A44}$$

and dividing both sides by $V^C$ results in.

$$NC = \frac{N^N}{N^C} \left( 1 - \frac{b^C}{V^C} \right) + \frac{b^N}{V^C}. \tag{A45}$$

Straightforward algebra gives.

$$\frac{b^C}{V^C} = \frac{b^C}{V_{iso}^C} \frac{V_{iso}^C}{V^C} = \nu^C \frac{V_{iso}^C}{V^C} \tag{A46}$$

$$\frac{b^N}{V^C} = \frac{b^N}{V_{iso}^N} \frac{V_{iso}^N}{V_{iso}^C} \frac{V_{iso}^C}{V^C} = \nu^C \frac{V_{iso}^C}{V^C}, \tag{A47}$$

and plugging *Equation A46; Equation A47* into *Equation A45* yields.

$$NC = \frac{N^N}{N^C}\left(1 - \nu^C \frac{V_{\text{iso}}^C}{V^C}\right) + \nu^N NC_{\text{iso}} \frac{V_{\text{iso}}^C}{V^C}.$$ (A48)

By (42), we have.

$$\frac{N^N}{N^C} = \frac{1 - \nu^N}{1 - \nu^C} NC_{\text{iso}},$$ (A49)

so that (*Equation A48*) may be rewritten as.

$$NC = NC_{\text{iso}}\left(\frac{1 - \nu^N}{1 - \nu^C} + \frac{V_{\text{iso}}^C}{V^C}\left(\nu^N - \nu^C \frac{1 - \nu^N}{1 - \nu^C}\right)\right).$$ (A50)

Adding and subtracting the term $NC_{\text{iso}}\left(\nu^N - \nu^C \frac{1 - \nu^N}{1 - \nu^C}\right)$ yields.

$$
\begin{aligned}
NC \quad &= NC_{\text{iso}}\left(\frac{1 - \nu^N}{1 - \nu^C} + \nu^N - \nu^C \frac{1 - \nu^N}{1 - \nu^C} + \left(\frac{V_{\text{iso}}^C}{V^C} - 1\right)\left(\nu^N - \nu^C \frac{1 - \nu^N}{1 - \nu^C}\right)\right) \\
&= NC_{\text{iso}}\left(1 + \left(\frac{V_{\text{iso}}^C}{V^C} - 1\right)\left(\nu^N - \nu^C \frac{1 - \nu^N}{1 - \nu^C}\right)\right) \\
&= NC_{\text{iso}}\left(1 + \left(\frac{V_{\text{iso}}^C}{V^C} - 1\right)\left(\frac{\nu^N - \nu^C}{1 - \nu^C}\right)\right).
\end{aligned}
$$ (A51)

Next, we invoke the hypothesis $b^N/b^C = V_{\text{iso}}^N/V_{\text{iso}}^C$, i.e. $\nu^N = \nu^C$. It follows from (*Equation A51*) that $NC = NC_{\text{iso}}$ independent of the external concentration and cell size $V^C$.

