## [Editor Report]

This work offers a simple explanation to a fundamental question in cell biology: what dictates the volume of a cell and of its nucleus, focusing on yeast cells. The central message is that all this can be explained by an osmotic equilibrium, using the classical Van't Hoff's Law. The novelty resides in an effort to provide actual numbers experimentally.

---

## [Decision Letter]

**Decision letter after peer review:**

Thank you for submitting your article "Control of nuclear size by osmotic forces in *Schizosaccharomyces pombeS. pombe*" for consideration by *eLife*. Your article has been reviewed by 3 peer reviewers, and the evaluation has been overseen by Naama Barkai as the Senior Editor and Reviewing Editor. The following individual involved in review of your submission has agreed to reveal their identity: Matthieu Piel (Reviewer #1).

The reviewers have discussed their reviews with one another, and the Reviewing Editor has drafted this to help you prepare a revised submission. As you will see, all reviewers support publications in principle but require revisions for the paper to be suitable for *eLife*. Please address all comments below.

*Reviewer #1 (Recommendations for the authors):*

My main recommendation is to discuss in more length the potential role of ions and other small osmolytes in setting the nuclear volume because of electro-neutrality and resulting electrostatic effects. The rest is just fantastic and I would not change the article very much.

It is important to mention that all these comments are the result of a detailed discussion with Romain Rollin, a PhD student with Jean-François Joanny and Pierre Sens, who convinced me from theoretical explanation and order of magnitudes that the electrostatic effects cannot be discarded to explain the volume of the nucleus. I had also discussed some of these points (without fully understanding them at that time) with Pierre Recho, resulting in the review of Cadart et al. in Nature Physics, 2019.

*Reviewer #2 (Recommendations for the authors):*

Some points to address:

A) GEMs:

In my view it is important to discuss the confusing aspects of data about the diffusion of GEMs in the nucleus and cytoplasm. The manuscript doesn't go into it as if assuming that this is not relevant to the discussion.

As examples of this:

On page 13 the paper draws a simple comparison between diffusive movement and crowding "less crowded at this scale". It should be made clear that this is a simplifying assumption. In fact, the anomalies in the comparison between data generated using GEMs and those using volumetric measurements suggest that the correlation between colloidal induced crowding that leads to osmotic pressure and diffusibility is far from perfect ­- given that the paper suggests that the nuclear and cytoplasmic compartments have a balanced osmotic colloid pressure and the same proportion of non-osmotically active material.

Figure 4B is a very strong result. However, 4B and 4D should be shown in the same way over the same range. This is important as it looks like behaviour is non-linear at high sorbitol concentrations in 4D. It would also be good to show a similar plot for 4B using the nuclear data.

I would suggest the axis in 4C be labelled "cell volume", since this measure has been assumed based on the idea that they act as perfect osmometers.

In calculating, authors say that they can use GEMs to match macromolecular concentration. However, this is different in nucleus and cytoplasm ­- when the previous model suggests this should be the same.

4C the axis "cell volume" is confusing. GEMs would be expected to diffuse similarly in large and small cells. In this case, cell volume refers to a change in colloidal osmotically active particles following an osmotic shock in a protoplast.

Better just to show 4B for both cells and protoplasts. As shown, it is hard to see in 4C how the fit is in cells treated with 1.0M sorbitol.

The final section on GEMs suggests that GEMs provide a good measure of crowding. However, I think this is a bit misleading as the paper used GEMs to test ideas of crowding and, even though the match is imperfect, assumes that they are related.

B. LMB treatment

In Figure 5B, based on my understanding of the simple colloid osmotic model, one would expect to see no change in the diffusion of GEMs in the nucleus or cytoplasm following LMB. This is because the instantaneous movement of water/ions should compensate for any change in the size of the compartment.

If anything, since cytoplasmic proteins are entering the nucleus and getting stuck there following LMB treatment, I would expect to see a shift in which the nuclear diffusion coefficient comes to resemble that of the cytoplasm. This would be easier to understand if the diffusion coefficients, not the ratios, were plotted in 5B. Then one could compare nuclear and cytoplasmic diffusion coefficients.

C. CHX

The CHX experiment provides a good control for the FITC staining and for GEM diffusion. However, how can there be a 30% decrease in protein concentration in cells without them undergoing a change in volume as expected under the model? Do cells compensate by pumping ions?

Note since cells appear to grow in this experiment, these data suggest that cells can balance nuclear and cell growth without protein synthesis ­- something that goes against the model proposed at the end of the paper.

Finally, if protein concentration is down following CHX treatment ­- this should be visible in a shift in the isotonic set point if this is dominated by colloidal osmotic forces. I think it would be great if that could be tested.

D. Asymmetric cell division

I like the idea of using an asymmetric cell division test of return to homeostasis (as has been done in animal cells). This experiment is used to suggest that nuclear size is set by there being a fixed proportion of osmotically active protein that ends up in the nucleus. This is a nice idea because it would explain how cells can maintain N/C homeostasis without the need for feedback and is similar to the adder models for cell size control. However, the authors don't test this model.

It would be good to include tests to distinguish the null model from models in which there is homeostatic regulation of nuclear/cytoplasmic scaling or to tone down the conclusions.

Tests:

i) It is important to show that cell and nuclear growth rates are independent of the N/C ratio ­- as suggested by the growth model.

ii) If the authors are correct, the speed at which cells restore a perturbed nucleus/cyt ratio will be related to growth rate. This could be plotted.

iii) G2 cells should correct their N/C ratio faster than G1 cells and cells unable to grow should not be able to restore a perturbation in the ratio.

iv) To get a clearer picture of the correction, it would be nice to plot the N/C ratio at birth versus N/C ratio before division to get a picture of how cells correct over an entire cycle.

E. Other things

i) At the start of the manuscript (pages 4-6) many of the definitions are poorly defined.

Bottom of page 4: not a good description of the way osmotic pressure is generated by proteins as described by Mitchison.

ii) Page 5 and 6: The authors state that ions and small molecules are not involved in the process. It should be made clear that the model assumes that ions are not involved (assuming that they partition freely and equally across the nuclear envelope) but does not show they are not involved.

iii) End of Page 5: The text conflates the non-osmotic component with macromolecular crowding. This is confusing. Macromolecular crowding is related to the forces generated by colloidal osmotic pressure.

iv) Figures 2A and 2B: It will be confusing to readers to switch axes labels calculating change versus Ciso.

v) Bottom of page 10 ­- instead of saying "loss of material" it'd be better to explain what's in the image.

v) Figure 3BD ­- it's hard to see the dotted grey line. I think it would be easier if separate plots for isotonic/shock data were added.

vi) Finally, the authors state that their model can be applied even to cases where there is no compartment boundary, e.g. in bacteria.

The size of the nucleus in this case is determined by the expansion of a defined boundary and depends on the regulated flow of macromolecules through a pore, such that a fixed proportion enters the nucleoplasm. Therefore, while the analogy is an interesting one, I am not sure this model can be usefully applied to explain nucleic scaling.

vii) I couldn't understand why the non-osmotically active content of cells doesn't change with osmotic shock when cytoplasm is diluted. Can you explain?

viii) What can we learn about free parameters in the Phillies experiment?

ix) Figure S2B ­- it seems that what is being measured is the sum intensity, since the Y axis is reported in total fluorescence. Is this correct? If the data hasn't been normalised for volume, why is mCrimson reporting on protein concentration?

x) In S7 with CHX nuclei change shape. It looks like the envelope is no longer under osmotic pressure and has excess membrane. Why then isn't there a change in N/C? How were nuclear volumes calculated in this case?

*Reviewer #3 (Recommendations for the authors):*

The questions 1 to 3 in the Public Review can be addressed by text change and/or new analysis and experiments.

Regarding point 4 from the Public Review, testing the relative fits of the two models on the current data should be a relatively straightforward analysis. The text should also be re-written to give a more accurate representation of the previous model.

Regarding point 5 from the Public Review, it may be interesting to test these mutants with hypo-osmotic shocks. The previous data suggest that nuclear volume can be forced out of its homeostatic range by deregulation of membrane amounts, but it is not clear to me how excessive nuclear membrane would lead to a change in volume (rather than just creating an irregular shape due to membrane folds) if the only contribution to nuclear size is osmotic regulation. This should at least be discussed.

---

## [Author Response]

Reviewer #1 (Recommendations for the authors):My main recommendation is to discuss in more length the potential role of ions and other small osmolytes in setting the nuclear volume because of electro-neutrality and resulting electrostatic effects. The rest is just fantastic and I would not change the article very much.

We address the primary comments of this reviewer by including more discussion of small osmolytes.

It is important to mention that all these comments are the result of a detailed discussion with Romain Rollin, a PhD student with Jean-François Joanny and Pierre Sens, who convinced me from theoretical explanation and order of magnitudes that the electrostatic effects cannot be discarded to explain the volume of the nucleus. I had also discussed some of these points (without fully understanding them at that time) with Pierre Recho, resulting in the review of Cadart et al. in Nature Physics, 2019.Reviewer #2 (Recommendations for the authors):Some points to address:A) GEMs:In my view it is important to discuss the confusing aspects of data about the diffusion of GEMs in the nucleus and cytoplasm. The manuscript doesn't go into it as if assuming that this is not relevant to the discussion.As examples of this:On page 13 the paper draws a simple comparison between diffusive movement and crowding "less crowded at this scale". It should be made clear that this is a simplifying assumption. In fact, the anomalies in the comparison between data generated using GEMs and those using volumetric measurements suggest that the correlation between colloidal induced crowding that leads to osmotic pressure and diffusibility is far from perfect ­- given that the paper suggests that the nuclear and cytoplasmic compartments have a balanced osmotic colloid pressure and the same proportion of non-osmotically active material.

We thank the reviewer for these insightful points, which we address by revisions in the text. We agree that small but robust differences between Diffusion coeff. in the nucleus and cytoplasm indicate that D_eff_ does not hold a perfect correlation with colloid osmotic pressure. We now state that this likely due to the different composition of the nucleoplasm and cytoplasm that modulate GEMs diffusion.

Figure 4B is a very strong result. However, 4B and 4D should be shown in the same way over the same range. This is important as it looks like behaviour is non-linear at high sorbitol concentrations in 4D. It would also be good to show a similar plot for 4B using the nuclear data.

We unfortunately do not have the data with the nuclear GEMs over the whole range of sorbitol conditions in protoplasts. We feel that for the purposes of this paper the large range of sorbitol concentrations up to 1M is adequate for our conclusion that cytoplasmic and nuclear properties behave in a similar manner.

I would suggest the axis in 4C be labelled "cell volume", since this measure has been assumed based on the idea that they act as perfect osmometers.

The axis in 4C is labelled cell volume. Note that for clarification, we added the term “(osmotic shocks)” in the x-axis to clarify this graph depicts how the changes in cells volume upon osmotic shocks relate the Deff.

In calculating, authors say that they can use GEMs to match macromolecular concentration. However, this is different in nucleus and cytoplasm ­- when the previous model suggests this should be the same.

GEMS diffusion is indeed compartment specific. We highlight this point by adding panel B in figure 4 (former Figure S4C) showing that Deff in nucleus and cytoplasm are slightly different. This may be due to compositional differences in the two compartments. We are careful to say that the absolute numbers cannot be used to compare the two compartments. However, our data in Figure 4D suggest that it is valid to compare the relative differences of GEMs with each compartment.

4C the axis "cell volume" is confusing. GEMs would be expected to diffuse similarly in large and small cells. In this case, cell volume refers to a change in colloidal osmotically active particles following an osmotic shock in a protoplast.Better just to show 4B for both cells and protoplasts. As shown, it is hard to see in 4C how the fit is in cells treated with 1.0M sorbitol.

Yes, this is not about differences in large and small cells. To clarify this point, we added the term “(osmotic shocks)” in the x-axis, as this graph depicts how the changes in cells volume upon osmotic shocks relate the Deff. We also added a color background (light blue and blue) onto the plot to show the hypotonic and hypertonic area.

The final section on GEMs suggests that GEMs provide a good measure of crowding. However, I think this is a bit misleading as the paper used GEMs to test ideas of crowding and, even though the match is imperfect, assumes that they are related.

We have clarified that the GEMs can be used to inform on relative changes within each compartment.

B. LMB treatmentIn Figure 5B, based on my understanding of the simple colloid osmotic model, one would expect to see no change in the diffusion of GEMs in the nucleus or cytoplasm following LMB. This is because the instantaneous movement of water/ions should compensate for any change in the size of the compartment.If anything, since cytoplasmic proteins are entering the nucleus and getting stuck there following LMB treatment, I would expect to see a shift in which the nuclear diffusion coefficient comes to resemble that of the cytoplasm. This would be easier to understand if the diffusion coefficients, not the ratios, were plotted in 5B. Then one could compare nuclear and cytoplasmic diffusion coefficients.

We thank the reviewer for these insightful comments. We have carefully reconsidered the interpretation of these experiments. We agree with this reviewer that we may expect to see little change in the concentrations of macromolecules because they will be compensated by changes in nuclear size at steady state. In support of this view, we add new data showing that total protein/RNA show little change with LMB, the nuclei behave like ideal osmometers and membrane tension remains low in response to LMB.

C. CHXThe CHX experiment provides a good control for the FITC staining and for GEM diffusion. However, how can there be a 30% decrease in protein concentration in cells without them undergoing a change in volume as expected under the model? Do cells compensate by pumping ions?Note since cells appear to grow in this experiment, these data suggest that cells can balance nuclear and cell growth without protein synthesis ­- something that goes against the model proposed at the end of the paper.

In our experiments, we used a low concentration of CHX that only partially inhibits protein synthesis and cell growth. Cells continue to grow, and biosynthesis still occurs. We do not claim here that cells grow without any protein synthesis. Cell volume is dictated not so much by protein concentration as concentration of smaller osmolytes.

Finally, if protein concentration is down following CHX treatment ­- this should be visible in a shift in the isotonic set point if this is dominated by colloidal osmotic forces. I think it would be great if that could be tested.

The isotonic set point is dictated by overall osmotic pressure, not just the pressure from proteins. Thus, as proteins only contribute a small subset of the total osmotic pressure, a 30% loss of proteins would not be expected to produce a measurable effect on this total pressure.

D. Asymmetric cell divisionI like the idea of using an asymmetric cell division test of return to homeostasis (as has been done in animal cells). This experiment is used to suggest that nuclear size is set by there being a fixed proportion of osmotically active protein that ends up in the nucleus. This is a nice idea because it would explain how cells can maintain N/C homeostasis without the need for feedback and is similar to the adder models for cell size control. However, the authors don't test this model.It would be good to include tests to distinguish the null model from models in which there is homeostatic regulation of nuclear/cytoplasmic scaling or to tone down the conclusions.

We have added new experiments that support our proposed model.

Tests:i) It is important to show that cell and nuclear growth rates are independent of the N/C ratio ­- as suggested by the growth model.

We added these new data to Figure 7—figure supplement 1: A. We show that there is no correlation of cell growth rate gc with initial N/C ratio. B. There is however an inverse relationship with nuclear growth rate with initial N/C ratio. Thus, the rate of nuclear growth, not cell growth, varies with the NC ratio. C. This nuclear growth relationship arises because the rate of nuclear growth is dependent on initial cell size (as predicted by exponential growth). For example, a cell with a large N/C ratio has a normal-sized nucleus and a small cell size, and hence this nucleus grows slower than a similarly-sized nucleus in a larger cell. Together, these findings support our proposed growth model.

ii) If the authors are correct, the speed at which cells restore a perturbed nucleus/cyt ratio will be related to growth rate. This could be plotted.

In a new experiment, we slowed down cell growth rate with low dose cycloheximide and found that the rate of cell growth, nuclear growth and NC ratio correction all slowed proportionally. These new data are included now in Figures 7 G-I.

iii) G2 cells should correct their N/C ratio faster than G1 cells and cells unable to grow should not be able to restore a perturbation in the ratio.

*S. pombe* cells do not really have a G1 period; these experiments are done in G2-phase cells.

iv) To get a clearer picture of the correction, it would be nice to plot the N/C ratio at birth versus N/C ratio before division to get a picture of how cells correct over an entire cycle.

We agree it would be good to follow N/C correction over entire cell cycles and even in lineages over multiple cell cycles. However, because of technical reasons (photobleaching and toxicity) we were only able to get robust 3D data for 40 min time periods, less than a whole cell cycle.

E. Other thingsi) At the start of the manuscript (pages 4-6) many of the definitions are poorly defined.Bottom of page 4: not a good description of the way osmotic pressure is generated by proteins as described by Mitchison.

We improved the description in the text.

ii) Page 5 and 6: The authors state that ions and small molecules are not involved in the process. It should be made clear that the model assumes that ions are not involved (assuming that they partition freely and equally across the nuclear envelope) but does not show they are not involved.

We improve the description in the text (see also response to reviewer 1 comments).

iii) End of Page 5: The text conflates the non-osmotic component with macromolecular crowding. This is confusing. Macromolecular crowding is related to the forces generated by colloidal osmotic pressure.

We improve the description in the text.

iv) Figures 2A and 2B: It will be confusing to readers to switch axes labels calculating change versus Ciso.

We have revised the labels for clarification.

v) Bottom of page 10 ­- instead of saying "loss of material" it'd be better to explain what's in the image.

We have clarified this in the main text. “[…] this effect was due to loss of a portion of the cytoplasm trapped in the remaining cell wall during the process of protoplasting (Figure S3A).”

v) Figure 3BD ­- it's hard to see the dotted grey line. I think it would be easier if separate plots for isotonic/shock data were added.

We labeled the shocks with a red line in the main figures and separated plots for isotonic/shocked were added in Figure 3—figure supplement 1.

vi) Finally, the authors state that their model can be applied even to cases where there is no compartment boundary, e.g. in bacteria.The size of the nucleus in this case is determined by the expansion of a defined boundary and depends on the regulated flow of macromolecules through a pore, such that a fixed proportion enters the nucleoplasm. Therefore, while the analogy is an interesting one, I am not sure this model can be usefully applied to explain nucleic scaling.

The analogy comes from the gel theory in uses a coefficient of partition between cytoplasm and nucleoplasm for solutes instead of a membrane tension. We will clarify in the text.

vii) I couldn't understand why the non-osmotically active content of cells doesn't change with osmotic shock when cytoplasm is diluted. Can you explain?

The non-osmotically active content can be considered as the volume of solid phase within the cell that is not penetrated by water. Molecules, ions, lipids present in the cell have an intrinsic volume, swelling the cell or shrinking the cell will not change their volume as long as there is no loss of matter.

viii) What can we learn about free parameters in the Phillies experiment?

This Phillies model is semi-phenomenologic and the meaning of the parameters is not clear. It is believed to be linked to non-specific interactions between the solvent and the solute and temperature.

ix) Figure S2B ­- it seems that what is being measured is the sum intensity, since the Y axis is reported in total fluorescence. Is this correct? If the data hasn't been normalised for volume, why is mCrimson reporting on protein concentration?

We used a single mid-focal plane image to measure concentration, not a sum intensity. We clarified how this fluorescence intensity was measured in the Methods: mCrimson concentration was measured as the mean fluorescence intensity of a ROI selected in a single mid focal plane of the cell, corrected by the mean fluorescence of the background. We assumed that the intensity of mCrimson reports its concentration and that the cytoplasmic concentration of mCrimson is homogeneous for a given cell.

x) In S7 with CHX nuclei change shape. It looks like the envelope is no longer under osmotic pressure and has excess membrane. Why then isn't there a change in N/C? How were nuclear volumes calculated in this case?

Nuclear volumes were calculated by Limeseq 3D volume measurements, as described in the methods. This method does not assume any particular geometry. We don't know why nuclear shape is altered, but there could be reasons unrelated to osmotic pressures.

Reviewer #3 (Recommendations for the authors):The questions 1 to 3 in the Public Review can be addressed by text change and/or new analysis and experiments.Regarding point 4 from the Public Review, testing the relative fits of the two models on the current data should be a relatively straightforward analysis. The text should also be re-written to give a more accurate representation of the previous model.

We clarified the model from Cantwell and Nurse (2019) in the new version of the manuscript.

Regarding point 5 from the Public Review, it may be interesting to test these mutants with hypo-osmotic shocks. The previous data suggest that nuclear volume can be forced out of its homeostatic range by deregulation of membrane amounts, but it is not clear to me how excessive nuclear membrane would lead to a change in volume (rather than just creating an irregular shape due to membrane folds) if the only contribution to nuclear size is osmotic regulation. This should at least be discussed.

We are pursuing these hypoosmotic experiments currently, but we regard them as outside the scope of this paper. Our preliminary data show that the nuclei in lem2 lnp1 double mutants break in hypoosmotic shocks. The genes have been implicated in ESCRT dependent repair of the nuclear envelope, and so the cause of this breakage may be due to many factors.